# Frequencies and functions of vocalizations and gestures in the second year of life

Megan M. Burkhardt-Reed[1]*, Edina R. Bene[2], D. Kimbrough Oller[2,3,4]

1 Department of Communication, Social Sciences Division, University of California, Los Angeles, Los Angeles, California, United States of America, 2 Origin of Language Laboratories, School of Communication Sciences and Disorders, University of Memphis, Memphis, Tennessee, United States of America, 3 Institute for Intelligent Systems, University of Memphis, Memphis, Tennessee, United States of America, 4 Konrad Lorenz Institute for Evolution and Cognition Research, Klosterneuburg, Austria

☯ These authors contributed equally to this work.
* burkhardtreed@ucla.edu

**Data Availability Statement:** The data used for analyzing communicative events, illocutionary coding, and gaze directivity is available on Mendeley at https://data.mendeley.com/datasets/fk6gfr65p9/. The recordings cannot be made

## Abstract

Speculations on the evolution of language have invoked comparisons across human and non-human primate communication. While there is widespread support for the claim that gesture plays a central, perhaps a predominant role in early language development and that gesture played the foundational role in language evolution, much empirical information does not accord with the gestural claims. The present study follows up on our prior work that challenged the gestural theory of language development with longitudinal data showing early speech-like vocalizations occurred more than 5 times as often as gestures in the first year of life. Now we bring longitudinal data on the second year (13, 16 and 20 mo), showing again that vocalizations predominated, and especially in conventional (learned) communication; > 9 times more spoken words were observed than gestures that could be viewed as functionally equivalent to words (i.e., signs). Our observations also showed that about ¾ of gestures across these second-year data were deictics (primarily pointing and reaching), acts that while significant in supporting the establishment of referential vocabulary in both spoken and signed languages, are not signs, but have single universal deictic functions in the here and now. In contrast, words and signs, the primary semantic components of spoken and signed languages, are functionally flexible, making possible reference to abstractions that are not bound to any particular illocutionary force nor to the here and now.

## Introduction

### Vocalization and gesture in communicative evolution

Many have argued for a close relationship between gesture and language in terms of both evolution and development [1, 2]. A multitude of findings have been interpreted as showing that gestures are the first means that convey communicative intent prior to the onset of words around the first birthday [3–5]. During the second year, published empirical research has tended to suggest that typically developing (TD) children display a moderate preference for

publicly available due to the IRB conditions and participant permissions; parents did not give consent to share the information publicly. Please contact Ricky L.A. Tan, Associate Director, Research Compliance at the University of Memphis, 901.678.4272 or altan@memphis.edu, with data inquiries.

**Funding:** This work was supported by the Plough Foundation and grants from the National Institute of Deafness and Communication Disorders of the National Institutes of Health, R01DC011027 and R01DC006099, awarded to D. Kimbrough Oller. The funders had no role in study design, data collection and analysis, decision to publish, or preparation of the manuscript. https://www.nidcd.nih.gov/.

**Competing interests:** The authors have declared that no competing interests exist.

gestures over words [6–9]. In the context of these observations, it has come to be believed by many that infant gestures play a predominant role in the first human communication that presages language. But the matter remains debatable. Do gestures truly precede vocalization in modern human development and in the evolutionary origin of language? Or is vocalization more foundational for communication?

To the extent that the issue of language origins has been raised, most recent published opinions exploring evolutionary possibilities have leaned toward a gesture-first hypothesis [10–12]. Historically, studies that support the gestural origins viewpoint suggest that the widely acknowledged vocal limitations of non-human primates, compared to their greater gestural flexibility, offer relevant evidence [13–15]; the advocates of the gestural origins viewpoint reason that since we are primates, our most fundamental communicative inclinations should be expected to resemble those of other primates. Furthermore, there has been speculation that adults are better equipped to interpret a baby's nonverbal communications than verbal expressions prior to the appearance of intelligible speech [16]. However, it is not obvious that this is true. The opinion appears to depend on how one interprets the term "communication". If, for example, communication is defined to be limited to acts that designate objects or other entities (deictics) and thus supply a basis for reference to objects or other entities in the here and now, then indeed, early infants are hampered in the vocal domain by not having words with which to make reference to objects or other entities. Pointing and reaching, on the other hand, are gestural acts that can serve the deictic function at least by late in the first year.

But if we define communication more broadly to include, for example, affective displays, it can be seen that vocalizations serve such functions from as soon as infants can breathe [17], not just in crying and whimpering, but also in vast numbers of speech-like vocalizations ("protophones", [18]) that begin on the first day of life and greatly outnumber cries across the whole first year [19]. Furthermore both vocalizations and gestures can be viewed as fitness signals, especially if they are produced in comfort, and although infants do not have to intend such actions as communications, they may serve the function of fitness signaling even so on occasions when caregivers notice them [20, 21].

In fact, there has been widespread dispute regarding the gestural origin of language. With regard to non-human primates, for example, a review of extant literature found that primate gestures did not clearly show greater evidence of key language properties (such as intentionality or turn-taking) than vocalization [22]. Further, studies of early human development challenge the idea that gesture precedes vocalization in communication [23–25]. Key points are that vocal babbling appears to be a better predictor of first word development than pointing [26] and the raw amount of vocalization greatly exceeds that of gesture across the whole first year [24].

One might imagine that the default hypothesis for those unfamiliar with the gesture-first literature would be that vocalization would have played the primary role in the origin of language, simply because language is primarily vocal. There are many categories of vocalization that human caregivers recognize in early communication development, including sounds not closely related to language such as crying and laughter, along with several more which are termed, for example, "vowel-like" sounds (technically "vocants"), squeals, growls, and raspberries. Modern human infants display a drive to engage in vocalization, to listen to their own sounds and those of caregivers, and to pay attention to faces, especially the faces of mothers from birth [27, 28]. It has been reasoned that the massive amount of vocal exploration in human infants forms a foundation for language [29–31].

## Natural selection of communicative signals

These vocal tendencies of infants suggest humans have been naturally selected to display inclinations and capacities from early infancy that promote communication with their caregivers and presumably later with all the members of their group. The patterns are compatible with the essentials of evolutionary developmental biology (evo-devo), a modern advance in biological theory [32, 33]. Evo-devo emphasizes that major evolutionary changes have often involved selection on changes in rates and/or patterns of early development. Within the evo-devo perspective, it has been proposed that selection pressures that may have been imposed on human infants make them inclined to seek and construct vocal capacities through vocal exploration [18, 21]. Ultimately the exploration of vocalization in infancy forms a foundation for language, and the evo-devo perspective supports a constructionist view in which infants and young children are portrayed as building vocal language in large measure on their own [34].

The longitudinal observational literature on typical language acquisition indicates clear patterns in infants' early vocalizations and babbling, with an increase in not only the complexity of vocalization with age, but also of the degree to which pre-speech vocalizations resemble real words in natural languages [29, 35, 36]. That is, infants build the capability to produce speech-like vocalization and also to use it communicatively, increasingly interacting in a conversational and expressive manner with caregivers and gradually incorporating real words into the interactions. Thus, the rate and complexity of vocalization increases as children begin to use more fully semantic lexical items (words) even though they continue to produce non-speech protophones well into the second year [37].

Evo-devo offers a framework for understanding how vocal language may have evolved. The development of protophones progresses toward more and more speech-like vocalization, and the high activity level of human vocalization compared to the much lower level in other apes [38, 39] suggests there must have been strong selection pressure on that high vocal activity level throughout hominin history. The argument in favor of this idea invokes the notion that the hominin infant was born altricial (helpless) due to the narrowed hominin pelvis that accompanied bipedalism and thus required more long-term caregiver investment than in the cases of other ape infants [20, 21]. Consequently, it is argued that the hominin infant was under particularly strong selection pressure to produce fitness signals in the form of vocalizations suggestive of wellness, to augment the existing wellness indicators supplied by other factors such as normal motoric development, normal eating patterns, skin condition, and so on [40].

Vocalization is particularly well suited to serving as a fitness signal since caregivers do not have to be looking at infants in order to notice the extent to which protophones indicate comfort and/or well-being. Gestures, on the other hand, require caregivers to be looking if the actions are to serve as fitness signals. 4000 species of songbirds produce massive amounts of fitness signaling in vocal mating displays that are often broadcast even in the absence of any nearby potential mates [41], but birds use mate-attracting *visual* displays only when in the company of potential mating partners [42]. It is also notable that birds are born altricial, and that songbirds, like humans, produce a sort of babbling (subsong) in the fledgling stage [43], presumably both as a fitness signal and as a means of developing foundations for song, again in a manner that seems parallel to the human case.

Consequently, it is plausible that the human line has long been under strong selection pressure to use vocalization as a fitness signal, while other primates, being less altricial at birth, have experienced less such pressure and thus have not developed such a strong vocal tendency. Gesture is available as a flexible means of communication for both humans and other apes, but the requirement of visibility of the gesturer offers reason to doubt that selection pressure would have influenced gesture to constitute such an important fitness signal as vocalization

either in the origin of language or in the development of language. This reasoning is consistent with the observation that gestural activity occurs much less frequently than vocal activity throughout human life, except in cases of individuals who use sign language.

Nonetheless, there has been a great deal of attention paid to gesture in speculations about the origin of language. Theories of language origins in support of a vocal source are not absent, however [44, 45]. In contrast to previously held beliefs about primarily gestural flexibility in non-human primates, some recent studies in support of vocal origins actually indicate more flexibility in the vocal than gestural communication systems of non-human primates [46–48].

## Testing the origins question with quantitative developmental evidence

Our evo-devo reasoning has always included the following supposition: If the earliest language was gestural, it would be reasonable to expect to see extensive gesture usage in early development. In turn, if the earliest language was vocal, it would be reasonable to expect to see extensive vocalization in early development. We pursued this reasoning in our previous work [24], hereafter, "BR2021".

A key factor in our research involved developing a framework of description that would allow justifiable quantitative comparisons of rates of vocalization and gesture in the first year. We view this as an important step and, according to recent reviews, clear definitions and criteria for classification have not been a major focus of empirical investigation on gesture and vocalization as language foundations in prior research [49, 50]. A key issue in developing an appropriate framework of description is recognizing the special role played by gestural transmission of deixis (e.g., pointing and reaching), a particular communicative function that is not easy to transmit in vocalization. These prelinguistic deictic gestures appear to be universal, having been documented across a variety of cultural groups, with essentially spontaneous emergence of reaching early in the first year and of pointing around 9 months [51–54]. Studies have shown that children increase their deictic gesture use, specifically pointing, across the second year as a foundation for and an accompaniment to their emerging lexicons [55–58]. One reason deictic gestures seem so important is that they can designate a referent visible to both speaker and hearer, and discerning their message may thus be less cognitively demanding than *conventional* gestures that have to be learned and may require more inference on the part of the observer (e.g., iconic wiggling the body back and forth to suggest a worm) [33].

In BR2021 we argued that it is critical to recognize the distinction between the Universal deictic gestures as opposed to both Non-social gestures (e.g., rhythmic hand banging, opening and closing a hand as if practicing the movement) and Conventional gestures, which must be learned (such as performative clapping or waving hello along with symbolic gestures that name entities or actions). The BR2021 paper advanced the claim that these three types of gesture should be consistently distinguished, providing a basis for comparison with three parallel kinds of vocalizations: Non-social vocalizations (protophones that are not directed to others), Universal vocal affective displays (such as crying and laughter) along with Universal protoconversation and affective expression with protophones, and Conventional vocalizations (both performative and symbolic words). We specifically noted that the Universal types are neatly distinct across the modalities. Deixis can be performed with ease in the gestural domain, whereas affective communication cannot. Affect can be communicated by gesture only after lexicalization of gestures as signs. In contrast, affective communication occurs in vocalization from the first day of life (e.g., in crying) and also in affectively communicative protophones, but deixis can only be performed vocally after notable learning of speech ("look at the bunny that just hopped into my office"). Of course, affect can be communicated by facial expressions from the first day of life.

In the present study, we continue the work of BR2021 into the second year, where reaching and pointing are expected to play significant roles, and some signed language symbology is expected also to emerge even in infants where sign language is not present in the household. At the same time, vocal communication is expected to grow rapidly. Thus, we aim to determine again the quantitative extent to which gesture and vocalization differ, this time in the second year of life. In addition, the effort affords the opportunity to add to evidence regarding the likelihood that language originated primarily in either the gestural or the vocal domain in our ancient ancestors.

### Study goals and hypotheses

Our primary goal is to extend the research initiated in BR2021 into the second year, comparing voice and gesture rates by assessing non-social usage, lexicality, universality and directivity of vocal and gestural communicative behaviors. To explore meaningful patterns of similarity and distinction between gesture and vocalization, the present study observed typically developing children at 13, 16, and 20 months in naturalistic laboratory recordings to evaluate the following hypotheses:

1. BR2021indicated higher rates of vocalizations compared to gestures in early communication development. Thus, we hypothesize that vocalizations will occur at a higher rate than gestures overall in the second year. More specifically:

   a. *Non-social* vocalizations will occur more frequently than *Non-social* gestures.

   b. *Universal* vocalizations will occur more frequently than *Universal* gestures.

   c. *Conventional* vocalizations will occur more frequently than *Conventional* gestures.

2. Based on data from BR2021, gaze toward another person appears to occur more during vocalization in the first year than during gesture. But the pattern may change in the second year because as gestures (particularly pointing and reaching) become more prominent as intentional communications in the second year, the infant may need to ensure that caregivers are looking, in order to make the gestures communicatively effective. We project the following age effects in the second year:

   a. At 13 months vocalizations will occur more frequently with directed gaze than gestures.

   b. At 16 months vocalizations will occur more frequently with directed gaze than gestures.

   c. At 20 months gestures will occur more frequently with directed gaze than vocalizations.

## Materials and methods

### Participants

Approval for the research was obtained from the IRB of the University of Memphis. Data were acquired from the University of Memphis Origin of Language Laboratories (OLL) archived longitudinal audio-video recordings. Parents of all participants provided written consent for the longitudinal study. For the present study, we selected available recordings of 12 parent-infant pairs (6 male, 6 female) in the second year. The pairs were recorded while engaged in naturalistic play and interactions in the OLL. Families were recruited from child-birth education classes and by word of mouth to parents or prospective parents of newborn infants. Interested families completed a detailed informed consent indicating their interest and willingness

to participate in a longitudinal study on infant sounds and parent-child interaction. All families lived in and around Memphis, Tennessee, and all infants were exposed to an English-only environment. Criteria for inclusion of infant participants included a lack of impairment of hearing, vision, language, or other developmental disorders at the time of recruitment. Demographics and recording ages for each infant at each recording session are provided in Table 1.

All the infants passed a hearing test with Visually Reinforced Audiometry at the Memphis Speech and Hearing Center that had been scheduled for one of their regular longitudinal visits between 8 and 17 months of age. One infant that had participated in the longitudinal study was not included in the research because of a later diagnosis of ASD, although there was no indication of risk at the time of the original recruitment. Two infants had significant difficulties with otitis media during the period of the longitudinal study, resulting in ear tubes being installed by their otolaryngologists. Both these infants showed relatively low rates of gesture and vocalization compared to the group as a whole (0.35 and 0.5 standard deviations below the mean for gesture and 0.42 and 0.81 standard deviations below the mean for vocalization). Two additional infants showed language development patterns that were deemed by speech-language pathologists to be slow and both participated in speech therapy sessions after 24 months of age. One of these showed relatively high rates of gesture (0.5 standard deviations above the mean) but slightly low rates of vocalization (0.1 standard deviations below the mean). The other showed relatively low rates on both (1.2 standard deviations below the mean for gesture and 0.79 standard deviations below for vocalization).

## Laboratory recordings

Each of the longitudinal recordings included three sessions, each approximately 20-minutes in length, usually drawn from a continuous $\sim$60-minute recording. For the current work, we analyzed only the sessions termed "interactive" at approximately 13, 16, and 20 months. In the other two sessions of the 60 min, the caregiver was in the room with the infant either reading in one case or talking to an interviewer in the other. From these recordings, one interactive session was selected for each infant (total: 12) at each age (total: 3), for a total of 36 sessions. Considerable amounts of vocalization ($>$ 4 protophones per min) have tended to occur in all

**Table 1. Demographics and recording ages.**

| Infant | Gender | Age at Recordings (months; weeks) | | |
|---|---|---|---|---|
| | | 13 mo. | 16 mo. | 20 mo. |
| 1 | F | 13;1 | 16;2 | 20;1 |
| 2 | M | 13;1 | 16;0 | 20;1 |
| 3 | M | 13;1 | 16;0 | 20;0 |
| 4 | F | 13;3 | 16;3 | 20;3 |
| 5 | F | 13;2 | 16;2 | 21;1 |
| 6 | F | 13;1 | 16;0 | 20;3 |
| 7 | F | 13;0 | 16;0 | 20;0 |
| 8 | M | 13;0 | 16;1 | 20;1 |
| 9 | F | 13;1 | 16;2 | 20;2 |
| 10 | M | 13;3 | 16;1 | 20;3 |
| 11 | M | 13;0 | 16;0 | 20;0 |
| 12 | M | 13;1 | 16;1 | 20;1 |
| Average age in months; weeks $M$ ($SD$) | | 13;1 (0;1) | 16;1 (0;1) | 20;2 (0;3) |

This table displays recording ages and demographics in months and weeks at each session for each participant.

three session types in OLL research (Oller, et al. 2021). We chose to analyze the interactive sessions, hoping to maximize the amount of observable gesture.

The laboratory setting was designed to resemble a child's playroom equipped with eight cameras positioned in the corners of the room. Both parents and children were provided with high fidelity microphones worn in an infant vest and on the parent's collar/shirt.

In the interactive sessions, the parents were instructed to interact naturally and playfully with the infant for the designated period of approximately 20 minutes. An experimenter in the adjacent control room selected two channels of video for recording at each point in time. The cameras were switched as needed to obtain a view of the child's face as well as another view of the interaction between the child and the parent and/or researchers during the recording.

The actual recording time for these interactive sessions averaged 20.5 min with a standard deviation of 1.4 min. Because of this variation, the counts of communicative events in both modalities were converted to rates per min for all data from each recording, and the per min data were entered into the GEE analyses reported in Results.

## Coding

All the coding was conducted in the same software environment used in BR2021 (AACT, Action Analysis Coding and Training [59]). The software provides simultaneous viewing of two channels of video and audio synchronized to the videos with frame accuracy. Real-time acoustic displays are provided in TF32 [60]. More detailed information about the coding software environment can be obtained from previous work [61].

The present research aimed to examine the frequency and directivity of communicative acts in the second year for both gesture and vocalization. Consequently, coding for the primary data collection was conducted in a way similar to that of BR2021, with both gestures and vocalizations coded on separate passes using repeat-observation coding (repeat viewing of two synchronized video channels for gestural type and repeat listening for vocal type). This methodology allowed us to flexibly implement our descriptive framework for maximally meaningful comparison between vocal and gestural events. As in the prior study, we drew a distinction between actions (observable behaviors such as pointing, producing a protophone, and so on), and functions, where functions indicated potential communicative illocutionary *intent* (designation, conversation, refusal, naming, etc.). The dimensions (or "fields") of coding were vocal actions, gestural actions, gestural illocutionary functions, vocal illocutionary functions, gaze directivity during gesture, and gaze directivity during vocalization.

The first author was the primary coder. Before coding a segment, she viewed each approximately 20-minute recording to gain perspective on the flow of events from both infants and caregivers. Vocalizations and gestures were coded for each segment in two separate passes. In the first gesture pass, the coder used repeat-observation to designate events and create boundaries for individual gestural events. Repeat observation allows coders to determine onset and offset boundaries of each vocal and gestural event. AACT allows coders to create boundaries by placing a cursor at the start and another cursor at the end of each event. Then, a label for the event can be selected from a coding panel or by a designated keystroke. All vocal acts for each of the 12 participants had already been coded in a previous study. Consequently, we used the boundaries from the previous vocal act coding for the present study.

Once an event is bounded, a placeholder can be created in a new dimension ("field") and the sequence designated by the placeholder can be selected and played repeatedly in AACT. In the second pass, the coder used the boundaries for each action in the vocal and gestural domain to automatically create placeholders for a new illocutionary dimension (for definitions of illocutionary forces, see [21, 24, 62]), the categories of which indicated the social or non-

social functions of the gesture or vocalization. The bounded time frames were transferred to a new coding panel without showing any vocal or gestural action codes, which allowed the primary coder to designate an illocutionary function for each action in each modality while being blind to any previously designated codes. In the final step prior to analysis, we collapsed observed vocal and gestural functions into the three global function types, *Non-social*, *Universal*, *and Conventional*, to allow theoretically-motivated comparison across the modalities. A full list of the illocutionary functions utilized in the present work is provided along with the correspondence to gestural and vocal acts in Tables 2 and 3 respectively.

In a third coding pass, infant gaze directivity was coded during each vocalization and during each gesture, for all cases where at least one of the two video views allowed an infant gaze directivity judgement. Any instances where a judgment could not be made were coded as "can't see". The gaze coding determined whether a vocalization or a gesture was produced while looking at another person (with at least a glance) or while not looking at another person during the period of the vocalization or gesture.

**Global types of illocutionary functions.** Non-social acts, in accord with our definition, have no inherent social communicative function, although they can be brought into the service of communication (especially through learned associations with meanings) because they can be produced voluntarily. *Non-social gestures* such as rhythmic hand banging, the most common Non-social gesture observed in our work [63, 64], can be viewed as comparable to

**Table 2. Gestural illocutionary functions.**

| Function Category | Gestural Function | Definition |
|---|---|---|
| Non-Social | Non-social act | Any non-utilitarian act, not conveying communicative intent (e.g., rhythmic hand banging, body rocking) |
| Universal Social | Request object | Show desire to obtain something |
| | Accept | Receive something offered |
| | Offer | Present something to someone to accept/reject |
| | Request up | Show desire to be held or picked up (e.g., arms reaching up to indicate request) |
| | Designate | Indicate person or object of interest (e.g., pointing) |
| | Request help | Showing or bringing attention to a desired object or outcome (e.g., winding a toy) |
| | Refuse | Indicate unwillingness to do something (e.g., a hand block or turning away) |
| | Show | Make something visible to be perceived by another |
| | Seek attention | Show desire to engage with another (e.g., touching or tapping a person to engage) |
| Conventional | Conventional gesture | Displaying performatives (e.g. blowing a kiss) |
| | Social Playful | Engage in interactive exchange, usually involving an object (e.g., playing catch) |
| | Exult | Show happiness or excitement, celebrate (e.g., clapping hands together) |
| | Bye-bye | Wave "bye-bye" |
| | Greet | Wave "hello" |
| | Surprise | Gestural indicator showing excitement for an event or outcome |
| | All gone | Both palms up indicating that something is completely finished or used up |
| | Imitation | Gestural imitation immediately or nearly so after another's gestural production |
| | Signed name | Indicating reference to a particular object or person (e.g. signing for "hat" by patting head) |
| | Signed Comment | Indicating reference by providing signed information about the object or person |
| | Signed Sentence | More than one sign to indicate or convey information |
| | Continue | Responding in conversation with sign |

This list is intended to include illocutionary functions of gestures in infancy and early childhood that could conceivably be interpreted as gestures by any communicative partner or observer. The list includes the complete set used in this study. The terms are drawn partly from literature on ape and human infant gesture. The definitions of the Conventional gestures provide examples that we observed, but of course, there are culture or family-specific options for conventional gestures just as there are culture or family-specific options for conventional vocalizations.

**Table 3. Vocal illocutionary functions.**

| Function Category | Vocal Function | Definition |
|---|---|---|
| Non-Social | No Force | No discernable illocutionary intent |
| | Vocal play | Playing with sound (e.g. canonical babbling) |
| Universal Social | Complain | Displaying a range of distress sounds to indicate discontent |
| | Exult | Displaying laughter or other high arousal sounds to indicate joy or excitement |
| | Call initiation | Call for the attention of another person to start interaction or communication |
| | Continue | Continue a conversation using a protophone without active engagement or elicitation (e.g., mom talks to me and I respond) |
| | Imitation | Sound imitation shortly after another's vocalization |
| Conventional | Performative | Expression that serves as a performance |
| | Social Playful | Vocally displaying characteristics of objects or animals (e.g., car sounds "vroom" or mooing for a cow) during playful interaction |
| | Request | Asking for something using a word |
| | Offer | Presenting something to someone to accept/reject using a word |
| | Refuse | Indicate or show unwillingness to do something using a word |
| | Accept | Receiving something offered using a word |
| | Designate | Indicate a particular person or thing to share interest with another person using a word |
| | Show | Indicate a desire to share attention on an object using a word |
| | Name | Naming an object or person |
| | Comment | Comment about an object, entity, quality, or situation using a word |
| | Solicit | Indicating a desire to engage with or obtain something from someone using a word |
| | Question | Requesting for information using a word(s) (e.g., "where is it?") |
| | Answer | Responding to a question posed by another person using a word(s) |
| | Agree | Indicating a person or object of interest using a word |
| | Deny | Refusal to admit that something is true using a word |
| | Vocal Sentence | Expressing a complete thought using a word (e.g., "soft teddy" or "Oh, I see it!") |
| | Performative Request | Expressing desire for assistance using a word (e.g., "help") |
| | Bye-bye | Indicating that someone or something is leaving or no longer in sight using a word |
| | Uh-oh | Indicating a mistake or that something bad happened |
| | All gone | Indicating that something is completely finished or used up |
| | Wow | Expressing excitement and/or surprise |
| | No | Indicating unwillingness to do something or expression of dissent |
| | Yes | Indicating willingness to do something or expression of approval or agreement |
| | More | Indicating desire for an additional amount |
| | Hello | Expressing someone's or something's arrival |

This table displays vocal illocutionary functions used in this study. As in Table 2, this table displays a complete list of illocutionary functions that were used in our study and that were presumably interpretable by any communicative partner or observer.

protophones (especially to reduplicated canonical babbling). Similarly, any hand position of American sign language (e.g., the "F" position) can be produced as gestural babble. However, such gestural acts, being voluntarily produced, have the potential to be adapted through learning to be conventional signs. For example, rhythmic hand banging could be adapted to indicate "something repetitive and boring", and the F hand could be adapted to have a meaning such as OK.

*Non-social vocalizations* include all the non-word protophones (including both non-canonical and canonical babbling) produced, for example, in periods of vocal play. For the purposes

of the present work we treated all protophones that were not directed to an adult, even if they occurred during social interaction, as Non-social on the grounds that protophones appear not to be developed on the basis of caregiver input, but instead on the basis of endogenous infant vocal exploration [21]. Just as gestural acts such as the F hand or rhythmic hand banging can be adapted as learned conventional communications, so protophones (but especially canonical syllables) can also be adapted as learned communications, especially as words.

Of particular interest for our comparisons are *Universal* forms of vocalizations and gestures. Universal forms of gesture and vocalization are usually communicative but require no associative learning, and their intended functions are interpretable to potentially any communicative partner. *Universal gestures* include acts that appear to have inherently social communicative intent. These types of gestures have a heavily deictic function. The most common examples are pointing, showing an object, or reaching for an object that cannot be obtained by a child independently. Universal gestures are capable of transmitting certain critically important communicative functions such as refusal, request, and designation (pointing), and it appears they require no learning.

In the exclusively vocal domain, these deictic functions can only be transmitted through symbols/words ("look at the butterfly behind you" or "I don't want that"). In BR2021, we did not include *Universal vocalizations* (including cry and laughter, affectively positive or negative protophones, and conversational protophones) in our counts because we did not recognize the importance of comparing Universal gestures with Universal vocalizations at the time. After reflecting on the uniqueness of the Universal gestures and the fact that they have essentially fixed functions (usually associated with designation), we were reminded that crying, whimpering, and laughter (which express negative or positive affect) also have essentially fixed functions of affect expression. We were struck by the fact that in both the gestural and vocal domains, the universal functions were essentially modality-specific in early infancy; the functions of deixis could be served gesturally but not vocally, while the functions of affect expression could be served vocally but not gesturally. Thus it became clear that one of the most interesting possible comparisons of gestural and vocal communication in infancy involved universal acts whose functions were markedly modality specific.

We did not make a direct comparison in BR2021 of the amounts of Universal gesture and Universal vocalization. Further, such a comparison has been ignored in all other prior attempts to quantify the relative amounts of gestural and vocal communication in infancy, as far as we know.

The data from BR2021 showed growth in the amount of Universal gesture across the first year. In preparation for the present work, we estimated from first-year vocal coding for the 12 infants of the present study that Universal vocalizations, crying and laughter as well as affectively positive or negative protophones, occurred more frequently in interactive sessions than Universal gestures at the early age (3 mo), but that rates for the two modalities for Universal communications became more comparable beyond 6 mo. The clear trend in BR2021 showed Universal gestures growing in frequency across the first year. There is reason to expect that trend to continue through the second year as infants increasingly point and reach communicatively both during purely gestural and combined vocal/gestural communications.

An additional universal tendency in human infants is vocal responsivity in face-to-face interaction using protophones, a pattern that is reported to begin from the first month of life [27, 63]. As the infant matures through the first year, it appears that this interactive tendency with non-word vocalizations grows. This tendency toward "protoconversation" is strong even though most of the infant vocalizations that occur even during periods where caregivers attempt to elicit vocalization are not responsive (not conversational), but rather appear to constitute disengaged, endogenous vocal activity [64]. The fact, however, that many such

vocalizations do indeed occur during engagement with caregivers without qualifying as conventional (learned) vocalizations poses an additional issue for our proposed categorization of vocalizations and gestures. It seems best to treat these conversational protophones as expressions of socially-directed affect since they occur predominantly in affectively positive exchanges. Consequently, in the data presented below, conversational protophones (along with other protophones judged to be affectively positive or negative) will be treated as Universal vocalizations. It is notable that, in a sense, gestures such as pointing, serve a dual function of deixis and visually-based interaction with caregivers; so both some of the Universal gestures and some of the Universal vocalizations we coded can be viewed as having an important feature in common, namely that they both involve and promote engagement of infants with their caregivers.

Thus, we include Universal vocalizations in our proposed descriptive scheme in order to afford a comprehensive three-way comparison among communication types (i.e., Non-social, Universal, and Conventional) across the two modalities. Our rationale for advocating an explicit comparison between the gestural universal deictics and the vocal universal vocalizations expressing affect is that without a lexicon, affect is not transmissible in the gestural domain, and designation (deixis) is not transmissible in vocalization, which makes this comparison important for our understanding of the relative ways children utilize universal communication in vocal and gestural language development.

*Conventional gestures* are those that are learned, and at the beginning, produced with a single discernible communicative (performative) function, such as waving "hello" or "bye-bye", or hand clapping in celebration. In our study of the second year, Conventional gestures also included signed words (e.g., patting the head with a flat hand with the palm facing down to produce the sign for "hat"), which are lexical items that can reach full semantic status across development, by expressing different illocutionary functions on different occasions. We also kept track of signed sentences and conventional games involving gesture (e.g., peekaboo and patty cake). *Conventional vocalizations* include performative words with a single illocutionary force, such as bye-bye, and fully semantic lexical items such as "hat", "doggie", or "mama/mommy", as well as sentences and vocal games.

## Coder agreement and outcomes

Both the primary coder (first author) and the agreement coder (last author) had extensive experience in coding of infant vocal types (cry, laughter and both precanonical and canonical protophones), vocal illocutionary coding, infant gesture coding, gestural illocutionary coding, and gaze directivity coding. They designed the gesture coding scheme together over a period of three years of coding and recoding of audio-video samples prior to BR2021. For the current effort the only difference in the coding task was that second year of life data were involved rather than first year of life data. The coding scheme, however, and the three global function categories remained the same as in BR2021, with the exception of our having added Universal vocal types in the second year analyses.

For coding of the current data, the agreement coder followed the same coding procedure as the primary coder, using the criteria outlined in BR2021. Twenty-three segments from among the 36 recordings for the agreement study were semi-randomly selected, and one five-minute segment was selected from within each of 23 different recordings. These were coded by the agreement coder in random order, mixing vocal and illocutionary fields as well as ages and infants. Seven or eight samples came from each of the three ages and all 12 infants were represented in the agreement samples. Five hundred thirty-five gestural and vocal events were

coded by both coders of the agreement set. The agreement coder was, of course, blinded to the coding of the primary coder.

The correlational results for the agreement coding showed high values. For Gestural illocution coding, there were 28 possible codes, 17 of which actually appeared in the agreement coding. For the two coders across the 11 segments that were coded by both individuals the average correlation for number of gestures coded in each of the categories was $r = .95$, n = 11. For Vocal illocutionary coding, there were 36 possible codes, with again 17 vocal illocution codes that were actually observed to occur. The average correlation between the two coders was $r = .86$, n = 12. Infant gaze directivity toward a caregiver occurred infrequently during both gestures and vocalizations. The agreement data for directivity were collapsed across the gestures and vocalizations so that all 23 segments were represented in the average correlation across the segments between the number of events that were coded as being directed toward a caregiver by gaze, $r = .94$, n = 22.

Since the data were ultimately collapsed across the three global functional categories of events (Non-social, Universal, and Conventional), the most important agreement issue concerns those three rather than subcategories for which correlations are reported above. The data on distribution of codes across the agreement coders resembled each other substantially and also resembled the patterns for the entire data set as presented in the first section of Results, below. Fig 1 presents the comparison for the two coders. The patterns show considerable similarity in that both coders showed Universal gestures as by far the most frequent gestures and Conventional vocalizations being more frequent than Universal vocalizations, which in turn were more frequent than Non-social vocalizations. The difference between the coders on Universal vocalizations was accounted for predominantly by a tendency of the 2nd coder to categorize more vocalizations as conversational continuations than the 1st coder, who tended to treat the same utterances as Non-social. This is a common coding discrepancy because it is often ambiguous to observers (even with audio and video) whether an infant protophone is intended to be directed to a caregiver who is attempting to elicit conversation (Continue conversation, Universal vocalization) or whether the infant is in fact disengaged (Non-social). Kappa agreement for the gesture codes was .8, while it was .65 for the vocalization codes.

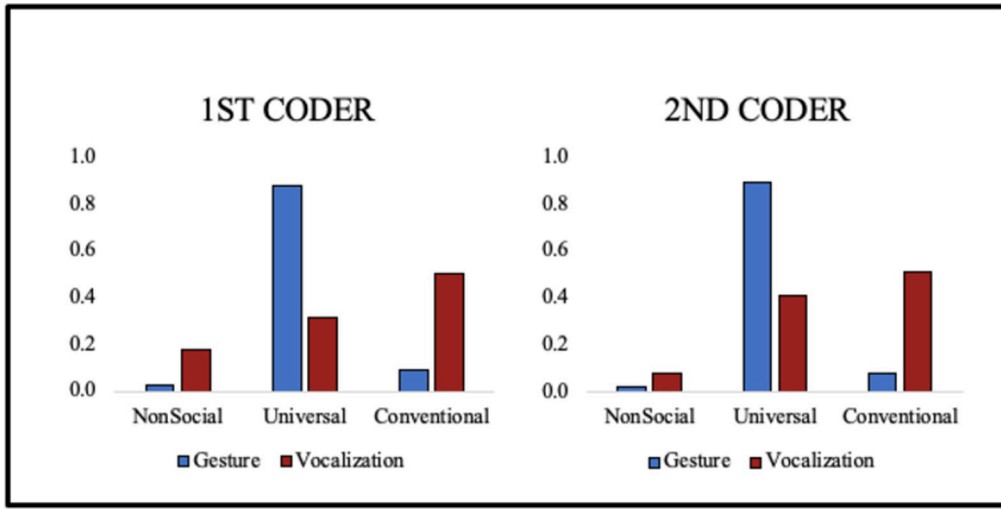

**Fig 1. Coder agreement data.** Proportion of events coded in the three global categories for the two individuals who independently coded 23 five-minute segments selected semi-randomly from 23 of the 36 recording sessions.

### Data analysis plan

Our method is observational rather than experimental. Consequently, we analyzed the data and present graphic representations where the primary issues at stake can be evaluated by overview of the figures. Still, the method allows formal statistical analysis. We have chosen to use Generalized Estimating Equations (GEE) [65], a non-parametric approach, for most of the formal analyses since the study is longitudinal, with potentially correlated participant data occurring at the three ages. GEE requires no independence assumption for the longitudinal data. Also our study has a small sample size, making a non-parametric method preferable [65, 66]. Also, GEE, unlike parametric approaches, requires no normality assumption, and the data in our study proved to show non-normal distributions. For the evaluation of the longitudinal hypotheses, therefore, we applied GEE analyses. Mann-Whitney U tests were utilized for formal comparison of gesture and vocalizations rates in the non-longitudinal summary data across the three global functional categories as presented in the next section.

## Results

### Overview

Fig 2 presents data on the relative rates of gesture and vocalization in the three global categories of illocution. More than twice as many vocalizations occurred as gestures (3276 vs. 1453), but the differences were dissimilar across the three global types. For Non-social events, vocalizations were > 5 times more frequent (652 vs. 114), while for Universal events, gestures were > 20% more frequent (1172 vs. 968). The greatest difference, as expected, was for Conventional events, where Conventional vocalizations were nearly 10 times more frequent than Conventional gestures (1656 vs. 167). Fig 2 presents gestures and vocalizations per minute across the three global types for the 12 infants. Vocalizations occurred most frequently as Conventional communications while gestures were produced most as Universal communications,

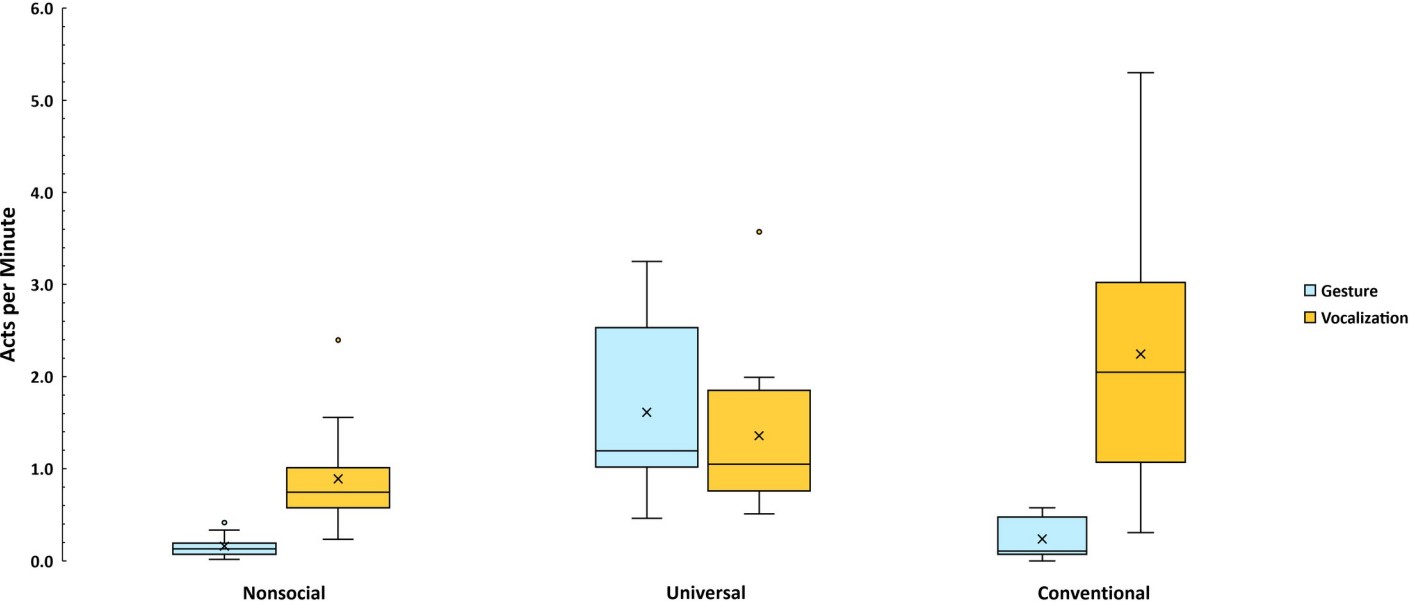

**Fig 2. Rate of communicative events per infant averaged across ages.** The box and whiskers plots represent all 4729 gestural and vocal events and are based on infant level calculations. Means are represented by an X in each box. In cases where outliers occurred, the upper whisker shows the highest value after outliers were removed from the distribution. All the raw individual data for the 12 infants are available in the data posted on-line.

with an average 10 Non-social and 14 Conventional gestures respectively for each infant, in contrast to 98 Universal gestures per infant. Vocalizations, on the other hand, were plentiful in all three global categories, with 54 per infant for Non-Social, 81 per infant for Universal, and 138 per infant for Conventional vocalizations. Mann-Whitney U tests indicated, in spite of large inter-subject variance, that there were significantly more Non-social vocalizations ($p <$ .001) than gestures and similarly that there were significantly more Conventional vocalizations ($p < .001$) than gestures.

**Hypothesis 1a.** The data in Fig 3 are broken down by age, where it is clear that Non-social vocalizations were considerably more frequent at all three ages than Non-social gestures, confirming our expectations. The GEE test of model effects showed no significant main effect of Age, but a significant interaction of Age with Modality (Wald $\chi^2$ = 28.8, df = 2, $p <$ .001) and a significant main effect of Modality (Wald $\chi^2$ = 14.3, df = 1, $p <$ .001). Fig 3 is not based on the GEE modeled data but on the original infant-level data in box and whiskers format just as in Fig 2. There were nearly 8 times more Non-social vocalizations than gestures at 13 and 20 months, and about 3 times more at 16 mo. The GEE Model results for each Hypothesis (1a-1c and 2) are provided in the S1 Appendix.

**Hypothesis 1b.** Fig 4 shows a complex pattern of interaction of Age with Modality on Universal gestures and vocalizations. Hypothesis 1b was not confirmed, because there was a weak tendency for there to be more Universal gestures than vocalizations. The GEE test of model effects revealed a significant Age by Modality interaction (Wald $\chi^2$ = 7.6, df = 2, $p =$ .022), reflecting the sharp difference at 13 months with respect to 20 months, where the former showed a higher rate of Universal vocalization ($>$ 60% higher than gesture) and the latter a higher rate of Universal gesture (more than twice as high as vocalization). Across the three ages, the Universal gesture rate rose while the Universal vocalization rate fell. The percentage of crying, whimpering, and laughing that entered into the Universal vocalization counts was low, only about 13%. Notably about 69% of those affectively valenced vocalizations were laughs. Thus, crying and whimpering contributed little to the patterns reported here.

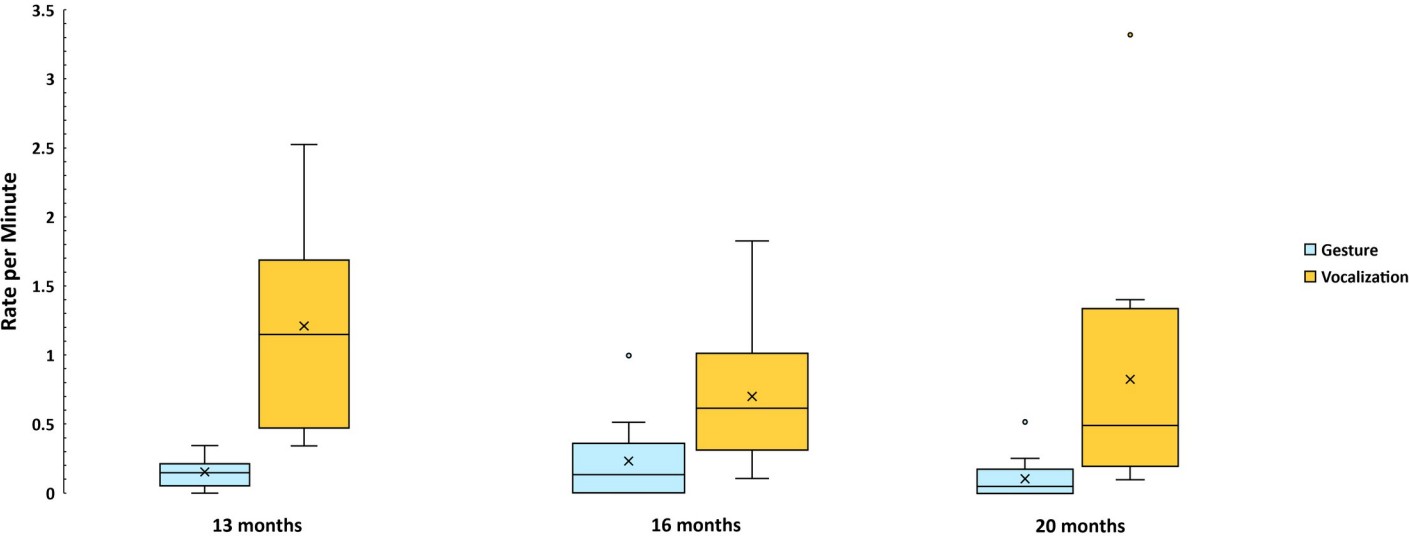

**Fig 3. Mean rate per infant of Nonsocial acts.** These data are based on the 766 gestural and vocal events that were coded as Non-social across three ages. The entire range of the communicative events per min is represented in each box and whisker plot. Whiskers are not visible in cases where the lower quartile value is equal to the minimum value, which is reflected in the box and whisker plots for 16 months and 20 months in this figure. The means are represented by X's in the boxes based on computation from the original data rather than from the modeled GEE values. The GEE analysis showed a statistically significant interaction of Age by Modality and a significant main effect of Modality, indicating higher rates of Non-social vocal than gestural events.

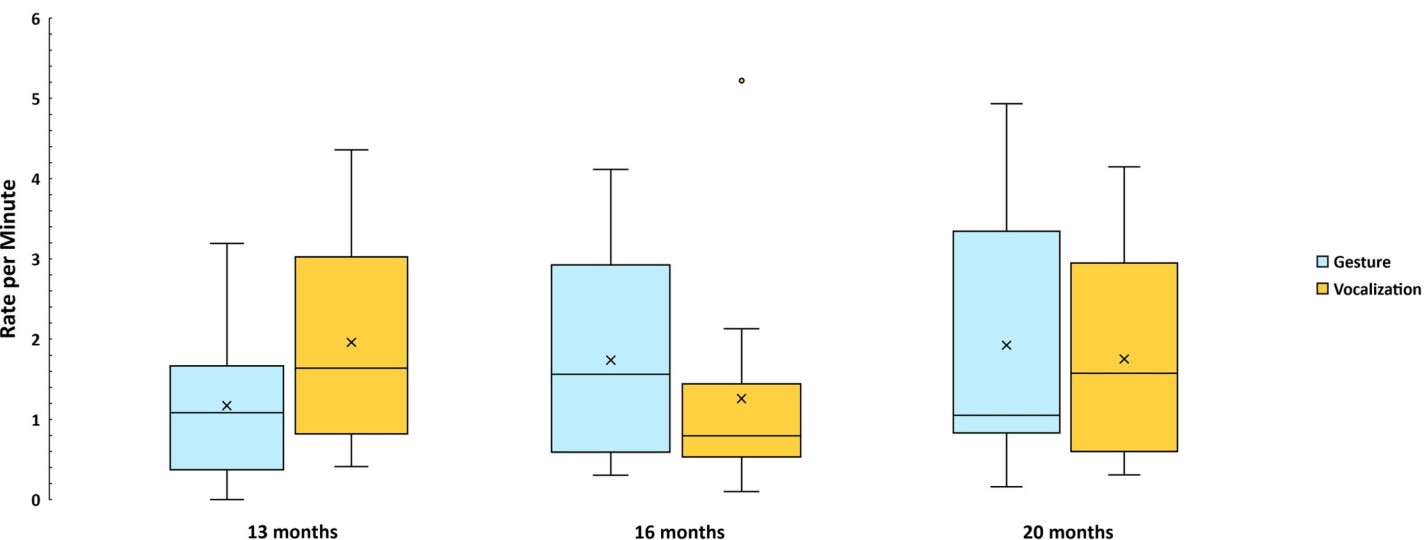

**Fig 4. Mean rate per infant of Universal acts.** The data are based on the 2140 gestural and vocal events that were coded as Universal communications across three ages. Again, the means of communicative events per min, represented by X's in the boxes, are based on computation from the original data rather than from the GEE values, which indicated a significant interaction of Age by Modality, reflecting the rise in Universal gestures across Age contrasting with the fall in Universal vocalizations across Age.

**Hypothesis 1c.** The strong tendency seen in Fig 5 for Conventional vocalizations (performative and symbolic words) to grow across Age is contrasted with a tendency for Conventional gestures (learned performative gestures and signs) to be very low across the entire Age range. This tendency was reflected in a highly significant Age by Modality interaction (Wald $\chi^2$ = 17.1, df = 2, $p < .001$) ) based on the GEE test of model effects, reflecting the fact that the difference between the rate of Conventional vocalizations and gestures grew dramatically from 13 to 20 months. There were $> 4$ times as many Conventional vocalizations as gestures at 13 months, $> 8$ times as many at 16 months, and $> 20$ times as many at 20 months. The analysis also showed statistically significant effects of Age (Wald $\chi^2$ = 9.7, df = 2, $p < .001$) and Modality (Wald $\chi^2$ = 30.3, df = 1, $p < .001$). Hypothesis 1c was thus strongly confirmed.

**Hypothesis 2.** The data on gaze directivity are presented also as box and whiskers plots in Fig 6, where the proportions of gaze directed communications per min across the modalities show that our expectation for gesture to be more gaze directed toward caregivers only at 20 months was violated. In fact, gestures were more gaze directed toward caregivers at all three Ages. The GEE test of model effects revealed a Modality main effect (Wald $\chi^2$ = 24.0, df = 1, $p < .001$), but no significant Age effect. The difference favoring gestures for directivity was smaller at 20 months than it was at 13 and 16 months. Gestures were accompanied by gaze directed toward a caregiver about 30% of the time at 13 and 16 mo, while vocalizations were accompanied by directed gaze about 24% of the time. In contrast, gestures were accompanied by gaze directed toward a caregiver more than 16% of the time at 20 mo, while vocalizations were accompanied by directed gaze about 15% of the time.

## Discussion

### General outcome summary

Our intent is to shed light on the origin of language, especially with regard to the relative roles of vocalization and gesture both in evolution and in development. The research reported here

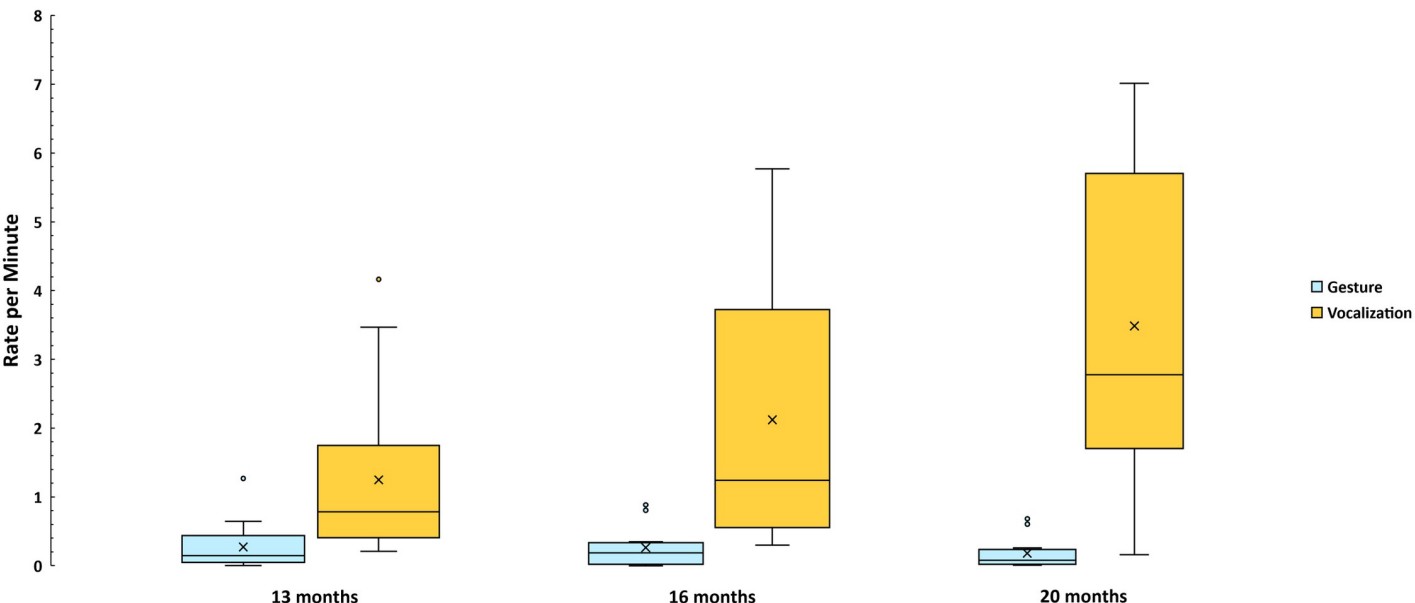

**Fig 5. Mean rate per infant of Conventional acts.** The data are based on the 1823 gestural and vocal events that were coded as Conventional communications across three ages. Again, the means, represented by X's, are based on computation from the original data on events per min rather than from the modeled GEE values, which indicated a significant interaction of Age by Modality, reflecting the rise in Conventional vocalizations across Age contrasting with no such rise in gestures across Age.

has shown that, as predicted in our reasoning, the bulk of communicative activity in the second year, as in the first, was vocal rather than gestural in the infants we tracked. Overall, more than twice as many vocalizations as gestures were coded across the 36 twenty-minute samples of caregiver-infant interaction in the second year of life, a difference that should be viewed in

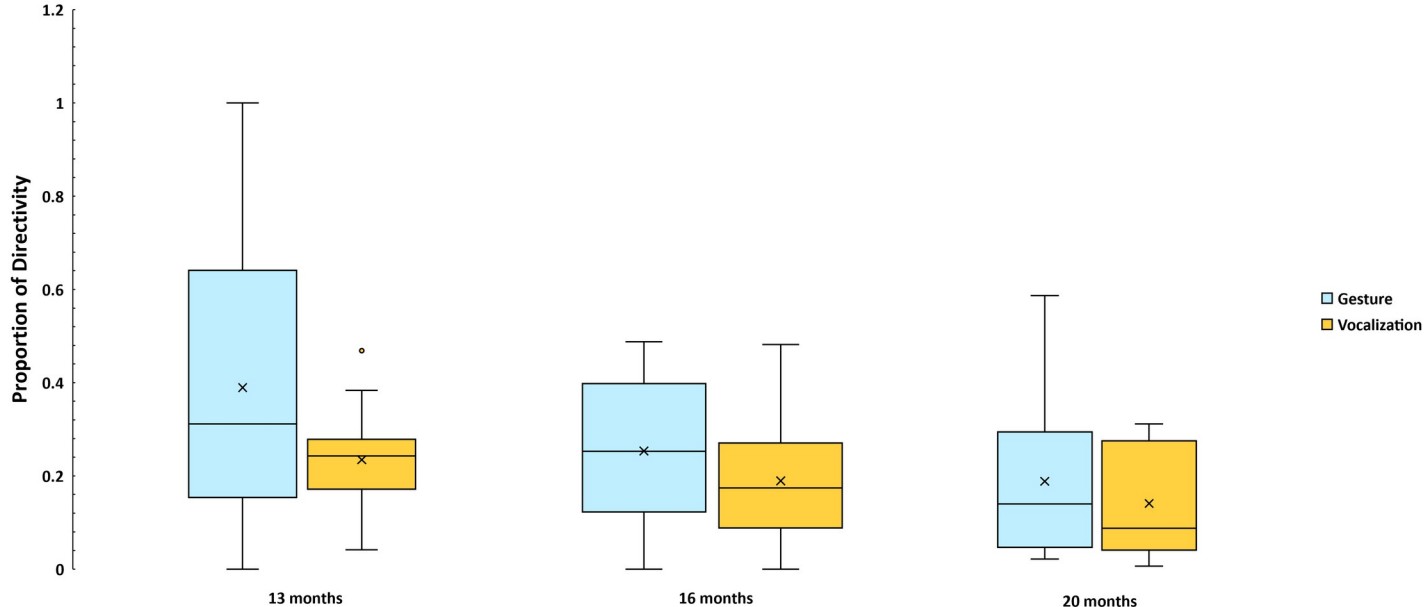

**Fig 6. Proportions of gaze directed gestural and vocal events.** The data are based on all coded gestural (1453) and vocal events (3253) where gaze direction could be judged based on at least one of the video signals. Again, the means reflect the raw data rather than the modeled GEE values.

light of the fact that across the first year in BR2021, 5 times more vocalizations occurred than gestures and that even at 11 months more than twice as many vocalizations occurred as gestures. But as we have been at pains to explain, the most insightful way to portray the relative roles of gesture and vocalization in language development is not through an overall comparison of rates, but through comparison of rates of communicative functions parceled into at least **three domains** reflecting different ways that gesture and vocalization can be used in infancy and also later in life.

Across the second year of life data, **Non-social** vocalizations, consisting of both precanonical protophones and canonical babbling, occurred > 5 times more commonly than Non-social gestures, the kinds of actions that might be thought of as signed babbling [67–69]. In contrast, **Universal** gestures outnumbered Universal vocalizations by about 20%, a fact that can be broken down into subcomponents: > 90% of the Universal gestures were points or reaches (the latter of which were largely interpreted by the coder as requests), with points occurring about 16% more frequently than reaches, whereas less than half the Universal vocalizations expressed positive or negative affect (a number that included a small number of laughs, cries, and whimpers along with a much larger number of protophones interpreted by the coders as transmitting positive or negative affect), the remainder having been interpreted as conversational continuations or conversational initiations using non-words, i.e., protophones. The biggest difference for the three global categories was found in that **Conventional** vocalizations (both performative and symbolic words and/or sentences) occurred > 9 times more frequently than Conventional gestures (performative or symbolic signs). This difference was greatly magnified at 20 months, where more than 20 times as many words per min occurred as signs (gestures that are functionally equivalent to words).

We conducted our study observing only circumstances where parents and infants were engaged in playful interaction. This approach was intended to maximize the amount of possible gestural interaction (although the amount of vocal interaction was also presumably enhanced), because a parent, even in the same room with an infant, but not engaged in communicating with the infant would likely not notice infant gestures. Vocalization produced in the same room has the advantage of being able to attract the attention of caregivers who are not communicatively engaged. It seems plausible that the greater amount of vocal than gestural communication in our study might be enhanced in the typical circumstance of the home, where caregivers have plenty to do other than engaging in playful interaction with their children.

## Biological perspectives on the origin of language

The outcome of our research is generally consistent with our perspective on the origin of language. We see considerable evidence that language is not only primarily vocal in mature users nowadays, but that communication is also primarily vocal at the beginning of human life, with primary evidence being the quantitative differences reported here. We further see reason based on these quantitative differences to support speculations that ancient hominins began in their evolution toward more complex communication than in other apes by evolving greater inclination and capacity for vocalization than in other apes [38, 39]. This evolutionary trajectory began, according to the reasoning, first with natural selection of exploratory infant vocalization as a fitness signal [20, 70]. Vocal fitness signaling laid groundwork for later selection of more complex uses of vocalization to express a vast array of possibilities involving both indefinitely large numbers of illocutions and semantic units (typically words) as well as combinations of these, making it possible to compose an indefinitely large set of possible sentences.

In principle, one might imagine that this kind of evolutionary trajectory might have begun with elaboration of gestural capabilities that were presumably already present in our distant

ape ancestors. But the developmental evidence does not show the traces we would expect if in fact gesture had formed the primary foundation for the massive differences that have been observed between the way humans communicate and the way our ape relatives communicate [39]. Gesture occurs extremely infrequently in human infants in the first three or four months according to the observations of BR2021. Yet, humans show massive amounts of protophone vocalization (4–5 utterances per min every waking hour) during the same period, and the tendency to produce protophones at very high rates even begins two months prior to due date in infants born prematurely and still in neonatal intensive care [17]. Our ape relatives, in contrast, show far less vocalization in infancy [39] and the vocalizations they do produce have never been reported to be purely exploratory, while exploratory vocalization is the hallmark of human infant vocalization throughout the first year [29]. So it makes sense to reason that a major transition must have occurred in ancient hominins, a transition where vocalization came to be produced far more frequently and far more flexibly than in other apes, and the fitness signaling theory suggests this transition occurred long before language in any modern sense existed.

Significant gestural usage does begin to occur more frequently in the second 6 months of life than the first (BR2021), but still far less frequently than communicative vocalization. The present data confirm that the first-year trend of increasing gestural usage (especially intentionally communicative gesture) also continues into the second year. Yet vocalization growth moves faster, and as we have seen, there is no point in time that we have observed, where gesture actually provides the quantitatively primary form of human communication. Vocalization appears to dominate at every point.

It might be suggested that the higher rate of vocalization than gesture in early life can be accounted for by the fact that fine motor control of the arms, hands and fingers does not develop until later than phonatory control. But the suggestion in fact merely restates the finding rather than explaining it. Yes, the fine motor control of protophones occurs earlier, but the appropriate question is why? We contend that an evolutionary explanation is needed and that the fitness signaling theory provides the necessary reasoning. In accord with the theory, it was and is because of the altriciality of infants in the human line (their slow and long-term development of head control, arm control, locomotion, and fine control of the extremities) that there has been selection pressure on them to signal their wellness effectively enough to elicit the long-term care they require because of altriciality. Phonation, according to the theory, long ago in hominin history became the target of that selection pressure, and in that way phonatory activity in humans became a precocial feature of hominins and thus laid a foundation for later developments leading to language. At a later point in evolution, complex gesture and ultimately sign language can be seen as having become, in accord with the reasoning, important secondary developments to the primary evolution of language in the vocal domain.

## Contrasting approaches to the study of language origins

So why does there exist a widespread belief that language originates developmentally in gesture? In the following section we explore ideas to help explain why our data appear to support primarily vocal origins of language and to contradict the widespread belief in primarily gestural origins. It appears that differences in how gestures and vocalizations are categorized in different realms of research probably form the primary basis for the differences in these perspectives.

Consider first the suggestion that gestures precede words in language development [2, 58, 59]. Of course some gestural activity does precede the first words produced by infants, but it is also true that a great deal of vocal communication precedes the gestural activity. And our data

suggest communicative vocalizations outnumber gestures at every age throughout early infancy and up through 20 months, where our current research has so far ended. Thus we view the suggestion that gestures precede words as misleading because gestures do not precede vocal communications and never actually exceed them.

One needs, we think, to take a broader view of communication in order to assess the relative roles of gesture and vocalization in language development. We propose different criteria for categorization of infant actions than in many prior studies supporting gestural origins. In one clear case, a study sought to determine gestural roots of vocal communication and compared vocal word usage with "action gesture" usage [5] based on a subtest from the MacArthur Bates Communicative Development Inventories (MB-CDI, [71]. This methodology uses parent report and provides no basis for differentiation among many of the "Actions with Objects" in terms of which ones should be treated as gestures and which ones should be seen as purely Utilitarian. The method seems to us to have yielded a misleading impression because, for example, in the paper's Fig 1, it appears that gestures outnumbered words substantially throughout the 8–16 month period. But this impression depends on the assumption that Actions with Objects are actually comparable to words, an unreasonable assumption.

In our own categorization, "action gesture" events on the MB-CDI such as drinking from a cup, using a spoon to eat, or opening a door are treated as *Utilitarian*, not gestural (see BR2021 for the technical definition). We agree with the authors of the cited paper that such actions are important developmentally, and they may even be important precursors to some aspects of communication. But they should not, in our opinion, be treated as gestures, and certainly should not be treated as equivalent to words. There are additional steps that an infant would need to take in order to convert such actions into gestures. Suppose an infant picks up a cup and *pretends* to drink from it: We would treat such movements when they first appear as pretend play, because they implement a Utilitarian action (still not gestural, but closer to gesture than an act of drinking from the cup) related directly to drinking from a cup. In a subsequent developmental step, such an action could become a sign with the meaning "drink" or "milk", but first it would have to be shown that the infant had come to use the pretend-play-movement independent of any real or toy cup *as a gesture or sign* rather than as a Utilitarian act of imagination. In our view there must be a detachment of the physical action from the Utilitarian action in order for an action on an object to be converted into an action without the object, thus achieving the genuine status of gesture. A further step of learning would be required to turn that gesture into a sign (referring for example to the conceptual category of cups). The key point here is to draw the distinction between things we do with our hands because we need to eat, drink and open doors, for example, and things we do with our hands that are communications by design or purpose (true gestures). Our observations, because they are based on naturalistic audio-video recordings, put us in a position to draw distinctions among the steps of development where a child's actions on an object or on an imagined object not actually in the child's hand can be made, and thus gestures can be differentiated from Utilitarian acts—the MB-CDI provides no such distinction. Consequently, we view the problem with prior literature using "action gestures" as in the cited paper as one of insufficiently differentiated categorization. The authors of the paper [5] appear to have understood the distinctions we have discussed here, but their figure was, in our opinion, easily misinterpreted to show that gestures precede words in development.

A more direct comparison would ask how many words and how many gestural signs occur at various points in development. We know of no prior literature that has actually counted the number of times in a given period a child produces a word as opposed to a gestural sign, nor has any prior work assessed the communicative intent of observed events within such a

comparison. Our report indicates there were massively more words than signs in the 36 caregiver-infant interactions we observed.

Perhaps even more important to illustrating our objections to much of the literature under discussion, gestural-origin advocates have routinely ignored vocalizations that are not yet words, assuming that gesture is a precursor to language but that vocalization (prior to the development of words) is not. For instance, protophones (non-word vocalizations) are not counted at all by the authors of the paper under discussion [5]. Yet protophones are clearly significantly communicative, not just as fitness signals, but as indicators of infant state, and perhaps most importantly as mechanisms by which infants and parents engage in social interaction, the very embodiment of primary intersubjectivity [72]. It seems clear that face-to-face vocal interaction (often termed protoconversation) beginning in the first months of life is a necessary foundation for secondary intersubjectivity (joint attention), which in turn appears to be a necessary foundation for word and sign learning [73–75].

Emphatically, we do not claim gesture is unimportant in language and language development. Especially we support the logically well-founded claim that joint attention plays a crucial grounding role in the learning of substantive vocabulary [76]. And we have no doubt that pointing is an important method helping to establish joint attention in infancy and early childhood [77, 78]. Other research also suggests that pointing both by infants and by caregivers can support word learning in the second year of life [79] and perhaps especially so as the end of the second year approaches [80].

But our categorization of pointing (and reaching, to the extent that reaching is viewed as an act of joint attention) as a deictic event, with only one Universal communicative function (to designate an object or entity in the here and now), undercuts (or at least severely limits) the suggestion that a point plus a word is akin to a sentence [2]. A point is not itself a symbol, and cannot become one without giving up its natural function of designating whatever is pointed at rather than being limited to designating a single concept. The importance of pointing as a mechanism of establishing and developing joint frames of reference [77] does not promote it out of the status of an illocutionarily fixed action (the function being designation) into the realm of symbolism, where words and signs must be illocutionarily free to transmit a vast array of functions (naming, denying, correcting, requesting, insulting, and so on). Otherwise, they would not qualify as words or signs. We do not intend to diminish the importance of pointing and naming or pointing and producing words, but it is confounding to interpret pointing as if it were a word or even as a sentential frame.

In BR2021 we noted that at 11 months the infants we studied showed remarkably little pointing, < 2% of observed gestures, whereas in the present data from the second year of life, pointing accounted for 41% of gestures over the whole second year and 59% at 20 mo, a huge change from the first year. Much of this tendency appeared to be associated with picture-book naming. The caregivers often chose a book from among the possible items for interaction (several options were always available) and engaged children in a naming game, and this parental choice occurred especially at the older ages. The use of pointing appears to be very circumstance specific, and it is hard to know how often caregivers engage infants in the home in naming games where pointing plays a major role. Of course, our evidence is consistent with the idea that early word learning can be supported by pointing—indeed word usage in our data increased by a factor of 2.8 from 13 to 20 mo., during which period pointing increased by a factor of 5.4. But which determined which? Perhaps pointing drove word learning, but it might also be claimed that pointing was supported by word usage, and indeed much of the pointing we observed occurred in circumstances where caregivers named a picture in a child's book ("where is the goat", for example), and the child followed by naming the picture and/or pointing to it.

Gestural interaction with caregivers in our data was overwhelmingly associated with pointing, reaching, and other deixis-oriented Universal gestures. We observed no cases where gestures that might be viewed as sign babbling (Non-social gestures) were adapted to sustained face-to-face interaction, while protophones were routinely utilized in face-to-face protoconversation. So all in all, it appears that previous authors supporting gestural origins of language based on developmental evidence have counted many items as gestures that we think should not be so counted, and have neglected to count as vocal communications the great bulk of the vocalizations that occur across the first year and a substantial proportion that occur in the second.

## Summary on gaze directivity

Our data on gaze directivity provide food for thought. It was clear from BR2021 that gaze directivity was limited in the first year both for vocal and gestural actions, although the proportion of vocal events that were directed by gaze to a caregiver was higher than for gesture. We interpreted this pattern as indicating that the gestural material we witnessed in the first year was only rarely intended as communication, since a gesture when no one is looking can hardly be communicative. Vocalization in contrast just needs to be loud enough to be heard to be communicative, one of the key facts that has always been adduced to support vocal origins of language. Yet, once a communicative frame is established or once it is merely understood by two parties that an interaction between them is underway, gaze directivity is not actually necessary for either vocalization or gesture to be communicative. The recordings we studied were overwhelmingly defined by precisely this sort of clear interactive frame. Both parent and infant knew they were interacting much of the time. Consequently perhaps they even more rarely used directed gaze in the second year of life data than infants had in the first year of life (BR2021). In fact, gestures were slightly more commonly gaze directed than vocalizations in the second year data, with the amount of gaze directivity falling to less than 20% of both gestural and vocal events by 20 months. The communications in both modalities could in the great majority of circumstances be interpreted as intentional communications regardless of the lack of eye contact.

## Limitations

In this section, we discuss provisos about the generalizability of the evidence we have presented here and in BR2021. First, our evidence in these two studies pertains to development only, and our reasoning about evolution of language requires bringing into the discussion very different bodies of research. Second, our developmental evidence has limits of generalizability because in households where deafness is involved among immediate family members, and thus where sign language may play a major role in communication, the patterns of usage of vocalization and gesture would seem likely to be notably different from those we found in BR2021 and in the present work because the families (infants, siblings, and caregivers) were hearing people. In households where sign language plays a major role, much of our perspective based on the present research might need to be substantially modified, and we have developed no empirical basis for that modification. Third, there may be notable cultural differences in how much gesture is utilized both in infancy and adulthood, even in hearing households. In particular, evidence has been presented to suggest that Italian is a language where much more gesture is utilized than in other European languages, and that infants learning the Italian language may also use gesture more frequently than infants learning other languages [81–83].

Still, the research we have conducted is surely relevant to the question of relative roles of vocalization and gesture in the origin of language. This contention is based in part on an

evolutionary developmental (evo-devo) perspective [84, 85], where evidence has been mounting for decades indicating that major innovations in evolution typically require selection on developmental processes. If a major transition in evolution is to occur, it will typically not only involve changes in development, but traces of the evolutionary process will be left in development. We have expanded this reasoning to encompass the case of language evolution, by suggesting that the relative roles of vocalization and gesture in modern human development, both in quantity of events and in order of appearance of events, may well reflect relative importance of vocalization and gesture in the process of emergence.

## Supporting information

**S1 Appendix. Generalized Estimating Equations (GEE) models for each hypothesis.**
(DOCX)

## Acknowledgments

We wish to thank the families in Memphis whose infants participated in this research.

## Author Contributions

**Conceptualization:** Megan M. Burkhardt-Reed, Edina R. Bene, D. Kimbrough Oller.

**Data curation:** Megan M. Burkhardt-Reed, Edina R. Bene.

**Formal analysis:** Megan M. Burkhardt-Reed.

**Investigation:** Megan M. Burkhardt-Reed.

**Methodology:** Megan M. Burkhardt-Reed, D. Kimbrough Oller.

**Project administration:** Edina R. Bene.

**Supervision:** D. Kimbrough Oller.

**Visualization:** Megan M. Burkhardt-Reed.

**Writing – original draft:** Megan M. Burkhardt-Reed, D. Kimbrough Oller.

**Writing – review & editing:** Megan M. Burkhardt-Reed, D. Kimbrough Oller.

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
