## [Decision Letter · Decision Letter 0]

17 Oct 2023

PONE-D-23-20708Frequencies and functions of vocalizations and gestures in the second year of lifePLOS ONE

Dear Dr. Burkhardt-Reed,

Thank you for submitting your manuscript to PLOS ONE. After careful consideration, we feel that it has merit but does not fully meet PLOS ONE’s publication criteria as it currently stands. Therefore, we invite you to submit a revised version of the manuscript that addresses the points raised during the review process.

Upon careful review of the text and the reviewers' comments, it is evident that it is a valuable addition to the field of research it pertains to and is potentially worthy of publication.You should focus on addressing the comments made by Rev. 1 regarding the introduction and taking into account the methodological issues raised by both reviewers. Additionally, the way the data is visualized needs to be improved. You are advised to refer to the Rev1 comments for more detailed contributions.

We look forward to receiving your revised manuscript.

Kind regards,

Elena Theodorou

Academic Editor

PLOS ONE

Journal Requirements:

Did you know that depositing data in a repository is associated with up to a 25% citation advantage (https://doi.org/10.1371/journal.pone.0230416)? If you’ve not already done so, consider depositing your raw data in a repository to ensure your work is read, appreciated and cited by the largest possible audience. You’ll also earn an Accessible Data icon on your published paper if you deposit your data in any participating repository (https://plos.org/open-science/open-data/#accessible-data).

We will update your Data Availability statement to reflect the information you provide in your cover letter."""

6. Please upload a copy of Figure 7, to which you refer in your text on page 30 (in PDF format). If the figure is no longer to be included as part of the submission please remove all reference to it within the text.

7. Please include a copy of Tables 1 to 3 which you refer to in your text on pages 18 and 21 (in PDF format).

**Additional Editor Comments: **

Reviewers' comments:

Reviewer 1

The manuscript consists of a timely study that challenges gesture-first theories of language. The paper begins with a review of the literature, suggesting that gesture-first theories are in the ascendency. The authors then argue that on the contrary, it is more plausible that humans had selection pressure to use vocalisations more (compared at least to other primates). The main hypothesis of the paper is that if earliest language was gestural – we should expect extensive gestural use early in development, whereas if vocal – we should expect extensive vocal use in development. Comparisons between gestures and vocalisation (on any metric) are difficult, so I appreciate the authors descriptive framework (universal, non-social and conventional vocalisations/gestures). The sample size is quite small, but the observation time is densely sampled (60 mins total per participant across three sessions), so this is still very informative. The results suggest that (a) non-social vocalisations are far more frequent than non-social gestures (which does not change across the second year of life), (b) neither universal vocalisations and gestures were more frequent across the second year of life, however the tendency was that vocalisations decreased across the year and gestures increased, (c) conventional vocalisations were more frequent and increased across the second year of life, while conventional gestures were infrequent and did not increase/decrease over time, and finally (d) all types of gestures were more likely to be directed to caregivers (as indexed by gaze) than vocalisations across the second year. The paper is well written and important. I agree with the authors that clearly the claim that early gesture use exceeds vocalisation use in human infancy is unfounded in light of this (and other data). I especially enjoyed the discussion of action gestures – and share the authors concern that “true gestures” are often confounded with actions (that are not gestures) simply because they could one day be ritualised into gestures, but that does not mean that they are true gestures when first used by infants! The same concern might also be applied to chimpanzee gesture coding schemes (which often include manual actions, e.g., splashing, as gestures…). I do however have a number of concerns about the manuscript in its current form, detailed below. I think that all the issues I raise are surmountable and very much look forward to reading a revised manuscript. Introduction and literature review There is a missing step in the introduction. I agree that there are proponents of gesture-first theories (and there is a sense at which many in the field simply uncritically appear to accept it as a premise), but there also vocal detractors (excuse the pun). It would be of benefit to the introduction to include an extra section detailing the body of opinion and evidence already amassed that disputes gesture-first theories (in the Vocalization and Gesture in Communicative Evolution section) before launching into the evolutionary argument (Natural Selection of Communicative Signals). For example, there was a review paper suggesting that great ape data simply doesn’t support a gesture-first or vocal-first origin story (Frohlich et al., 2019 https://doi.org/10.1111/brv.12535). On the developmental side of things, there are other papers that provide evidence that would already count against gesture-first theories (e.g., McGillion et al., https//doi.org/10.1111/cdev.12671 ; Donnellan et al., 2020 https://doi.org/10.1111/desc.12843; McCune et al., 2020 https://doi.org/10.1177/014272372096682; Werwach et al., 2021 https://doi.org/10.1016/j.infbeh.2021.10158). Inclusion of a review of the literature that already supports the authors main contention would strengthen the manuscript. Universal vocalisations I am unconvinced by the justification in the current manuscript that crying, whimpering and laughter should be included as universal vocalisations. Universal gestures are defined with respect to communicative intent, however it’s not clear that laughter, crying and whimpering necessarily are produced with communicative intent (though it’s possible – and of course whether deictic gestures are always produced with communicative intent is debatable too [examples of children pointing for themselves, see Delgado, Gómez, & Sarriá, 2011: https://doi.org/10.1016/j.jecp.2011.04.010]). I am much more convinced that non-word conversational protophones should be included as universal vocalisations – both these and deictic gestures promote engagement with caregivers (and more richly, could be intended for this purpose on the part of the infant – though this is speculative [but often a speculation accorded to gestures only in the literature…]). I don’t quite see the parallels between deictic gestures with affective vocalisations – I do take the authors point that they have a modality-specific function (expressing affect -which gestures can’t do without a lexicon), but that doesn’t seem a good enough reason to group them together, especially because they are also grouped with vocalisations that might not have a modality-specific function (e.g., the conversational protophones – whose function [promoting engagement] could also be achieved gesturally [e.g., through showing: holding up objects to caregivers face, or even pointing at caregivers]). I think as it stands, universal vocalisations should consist either of crying, whimpering and laughter (to maintain the modality-specific function element) or conversational protophones (to capture the engagement element). As is stands, the justification is clearer to me that it should just be the protophones. Measures Sessions are approximately 20 mins (not exactly 20 mins). This makes it hard to interpret individual differences across participants. How close to 20 minutes are the sessions? If within a few seconds, this is fine. However, if discrepancies are significant, behaviours should be standardized as rates per minute (or rate per 20 mins if preferred) instead of as a raw frequency. The analysis should then be re-run with these standardized measures. Reliability Coding The first paragraph of Coder Agreement and Outcomes section is unnecessary – this is simply reporting reliability statistics from a previous study – not the current study – and it’s not clear why this is relevant. I suggest simply omitting this. The final paragraph concerning categorisation of vocalization and gesture into the three functional categories requires more quantitative reliability metrics. I agree (by eyeballing the plot) that this probably shows pretty decent agreement, however, given the way in which the data was coded, it should be possible to calculate kappa reliability (if I have understood the coding procedure correctly). I.e., for the instances in which both coders agreed there was a vocalisation or a gesture – did they agree on the function (i.e., did they both select an illocution code that would be categorized as the same function?). Overall though, it should be noted that reliability correlations currently reported are solid (and my impression is that the kappas will equally demonstrate this) – I have no doubt that this coding has been conducted meticulously, but there do need to be missing metrics included to confirm this. Analysis GEE does seem an appropriate method for the data, and likewise Mann-Whitney tests are appropriate in my understanding. The GEE models obtained need to be included in the paper in full (if even simply in tables in an Appendix). Data visualisation Given that the authors rely heavily on the visualisation of the data to do much of the work of inference (and correctly so I think!), the visualisations need work. I have a few suggestions: Distributions of data: Ideally the distribution of the datapoints themselves needs to be included (not just 95% confidence intervals and a mean). Numerous ways exist to do this, for example violin plots, half violin plots, and/or plotting the individual data points. E.g., in figures 3, 4 and 5, individual points (representing a participant) could be plotted in red at each time point for vocalisations and in blue for gestures, with summary statistics (mean/CIs) overlaid, perhaps in tandem with violin plots or similar. Model predictions. Equally, if GEE was implemented using an R package, it may be possible to plot model predictions (perhaps overlaid) on the figures automatically – which would be very informative. Of course it is possible to do manually, but this does require expertise. This is becoming much more of a standard approach to visualising data (i.e., raw scores – as you have here, overlaid with model estimates). This would really support the authors inferences from the data and I would strongly encourage that they attempt this. This would also remove the need for caveats to be included in the figure captions about what the model shows compared to what the figure seems to show (i.e., the confidence intervals relating to the the raw data not the SE/confidence intervals of the model predictions). Figures 2 and 5. For hypothesis 2 a pie chart is not informative – as the data comes from individual participants. Please plot the data so that the range of proportions of gestures/vocalisations combined with eye gaze across participants is clear (i.e., in the same way as figures relating to hypothesis 1 would be fine, y axis being proportion). As currently plotted, the discrepancy between the data and the model results (especially with regard to 20m timepoints) is hard to interpret (i.e., why there is a main effect of modality, not an interaction with age). Likewise for figure 2, though there are error bars, perhaps consider plotting the individual data points with summary statistics overlaid as suggested above (i.e., not as a bar chart), so the distribution of individual participant data can be observed. Data availability Currently the data for the study is not available. I agree fully that videos cannot be shared. It is perfectly possible however to have the data worksheet mentioned in the availability statement already uploaded into the OSF for example (it can be a private submission), with an anonymized link provided only for the purpose of review. Minor comments Fig 3. - Typo on y axis label (“Frequancy”). Also figure 3 error bars do not look correct and it’s a screenshot containing “plot area” in a box over the middle. I have suggested more substantial changes to figures above. Typo – in the Results paragraph headed “Hypothesis 1a”, the text refers to Figure 5 – this should be figure 3? Please include in the limitations section that this is still a relatively small sample size (in terms of unique participants). Please include line numbers in the next draft to facilitate navigating the text and providing feedback. When discussing possibility of vocalization to serve function of communication very early in development (e.g., page 2) consider citing Papaeliou et al., 2002 (10.1044/1092-4388(2002/024)) and/or Esteve-Gibert & Prieto 2012 (https://doi.org/10.1017/S0305000912000359 ) as these seem relevant too. Note, I could not see tables in my copy of the manuscript. Whether this is an editorial mistake or submission mistake, I’m unsure (I contacted the journal to ask, but didn’t get a response before submitting my review to the deadline). I look forward to reviewing the content of the tables in the next round of review.

Reviewer 2

**INTRODUCTION**

The literature review is relevant. The paper is clearly written with bibliographic references on par with latest published research. On page 14 the authors provide a clear description of the different types of gestures. I was glad to this this section because it provided a clear pathway to the information following the introduction.

I suggest that the authors provide the term typically developing (experimental group) earlier in the paper.  

**METHODOLOGY**

I am not clear on how toddlers were examined to exclude developmental/language challenges.  Dis children have any episodes of otitis media? Where they excluded? I mention this because under such circumstances even children and toddlers with neurotypical development use more communicative gestures to compensate for reduced vocal output and compromised hearing sensitivity.

Is there a rater-to-rater reliability score?  

**RESULTS**

No comments. This is a qualitative examination and data are clearly presented in a via descriptive non-param stats.

**DISCUSSION**

The authors describe the important performance and provide adequate explanation of the findings.  The writing is clear and the manuscript is well-organized.

The authors provide adequate description of findings which are combined with plausible interpretations.

The novelty of the study rests on the fact that the findings contribute and enrich the  database related to vocalizations versus gestures.

Reviewer's Responses to Questions

**Comments to the Author**

1. Is the manuscript technically sound, and do the data support the conclusions?

Reviewer #1: Yes

Reviewer #2: Yes

2. Has the statistical analysis been performed appropriately and rigorously? 

Reviewer #1: Yes

Reviewer #2: Yes

3. Have the authors made all data underlying the findings in their manuscript fully available?

Reviewer #1: No

Reviewer #2: Yes

4. Is the manuscript presented in an intelligible fashion and written in standard English?

Reviewer #1: Yes

Reviewer #2: Yes

5. Review Comments to the Author

Reviewer #1: Please see attached comments (in pdf form). Overall, my review is positive, but there are revisions that need to be made in order to get the manuscript into publishable form. I look forward to the next round of review.

Reviewer #2: I would like to thank you for giving me the opportunity to review this interesting and well-written piece of work. As per your request, here is the review of PONE-D-23-20708 entitled Frequencies and functions of vocalizations and gestures in the second year of life”. My recommendation is “Acceptance with minor changes”. My comments are mostly related to methodological and stylistic issues. The paper focuses on the critical theme related to the importance of gestures vis a vis vocalization as early indications of the origins of language. It follows a previously conducted study, with the current paper adding and supporting evidence of their theoretical EVO-DEVO framework. The theoretical framework of observation highlights the fact that more than ¾ of gestures across these second-year data were deictics (e.g., pointing and reaching), acts that while significant in supporting the establishment of referential vocabulary in both spoken and signed languages, are not signs, but have single universal deictic functions in the here and now. In contrast, words and signs are functionally flexible, making possible reference to abstractions that are not bound to any “particular illocutionary force nor to the here and now”. This topic is long overdue given the scarcity of data available from such topics related to the origins of language.

6. PLOS authors have the option to publish the peer review history of their article (what does this mean?). If published, this will include your full peer review and any attached files.

Reviewer #1: No

Reviewer #2: No

---

## [Author Response · Author response to Decision Letter 0]

18 Feb 2024

All responses to specific reviewers and editor comments is provided in the attached "Response to Reviewers" document provided with our re-submission materials.

---

## [Decision Letter · Decision Letter 1]

28 May 2024

PONE-D-23-20708R1Frequencies and functions of vocalizations and gestures in the second year of lifePLOS ONE

Dear Dr. Burkhardt-Reed,

Thank you for submitting your manuscript to PLOS ONE. After careful consideration, we feel that it has merit but does not fully meet PLOS ONE’s publication criteria as it currently stands. Therefore, we invite you to submit a revised version of the manuscript that addresses the points raised during the review process.

We look forward to receiving your revised manuscript.

Kind regards,

Elena Theodorou

Academic Editor

PLOS ONE

Journal Requirements:

Reviewers' comments:

Reviewer's Responses to Questions

**Comments to the Author**

1. If the authors have adequately addressed your comments raised in a previous round of review and you feel that this manuscript is now acceptable for publication, you may indicate that here to bypass the “Comments to the Author” section, enter your conflict of interest statement in the “Confidential to Editor” section, and submit your "Accept" recommendation.

Reviewer #1: (No Response)

Reviewer #3: (No Response)

2. Is the manuscript technically sound, and do the data support the conclusions?

Reviewer #1: Yes

Reviewer #3: Partly

3. Has the statistical analysis been performed appropriately and rigorously? 

Reviewer #1: Yes

Reviewer #3: Yes

4. Have the authors made all data underlying the findings in their manuscript fully available?

Reviewer #1: Yes

Reviewer #3: Yes

5. Is the manuscript presented in an intelligible fashion and written in standard English?

Reviewer #1: Yes

Reviewer #3: Yes

6. Review Comments to the Author

Reviewer #1: Please see my attached comments for one final minor revision. Overall, again my review is very positive.

Reviewer #3: The paper in general is very well written. I do have a few considerations though, regarding methodology and the conclusions. My comments are shown below:

• Line 96: it has been reasoned that in the massive amounts of vocal explorations …

METHODOLOGY/RESULTS

• I have an issue with the inclusion of the category “social plauful” in Universal vocalizations. Although animal sounds or object sounds are echopoietic, they are still conventional across languages, so they are the product of learning/associations. I would have liked to know what the percentage of this category is in Universal Vocalizations across ages, and also I suspect that the fact that there is no significant difference between gestures and vocalizations in the Universal category is due to the fact that this subcategory was wrongly included in Universal instead of conventional. When this happens the difference should become evident.

• The fact that the two coders agree only at .8 and .65 regarding social/non-social interactions creates a problem. The percentage is very low. (line 523/524). This suggests highly ambiguous data and problematic categories. Can you please explain how you handled this problem?

• I cannot see any figures in the Revision 1 attachment. Seeing them in the initial submission, please add descriptions of the axes in all figures

DISCUSSION

• I also have an issue with the discussion about what constitutes a gesture and a general utilitarian movement. Also I would like to bring into this discussion the issue of development of fine motor skills which takes time for infants, even in the first year of life. In other words, gestures involve fine motor movements, which take longer to develop than gross motor movements and sounds (a bit like having protophones and language sounds). So although there are movements with the hands, they are not considered gestures, as the latter take long to be produced in a controlled manner.

• A final issue concerns a point acknowledged in the limitations section – there are no data from deaf children and this suggests that we cannot know whether the patterns observed are due to an evolutionary reason or whether a special emphasis on vocalizations is the result of communicative reinforcement from parent interactions throughout the first year. So if we are going to talk about an evolutionary perspective emerging from the current data, we need to account for the fact that parents respond to vocalizations from day 1 but not to gestures. So how do we know that the current data, and preference for vocalizations over gestures, are the result of a genetic/evolutionary basis of language and not just conditioning/reinforcement for sounds instead of gestures?

7. PLOS authors have the option to publish the peer review history of their article (what does this mean?). If published, this will include your full peer review and any attached files.

Reviewer #1: No

Reviewer #3: **Yes: **Loukia Taxitari

---

## [Author Response · Author response to Decision Letter 1]

24 Jun 2024

Response to reviewers is addressed in the document included in our resubmissions materials. We have also included the Author responses to the reviewer comments included below:

Reviewer #1: Please see my attached comments for one final minor revision. Overall, again my review is very positive.

Thanks for your comments and noticing the oddity in the figures. They have been corrected. We have provided an explanation for why the minimum whiskers cannot be displayed when the minimum is at zero. 

Reviewer #3: The paper in general is very well written. I do have a few considerations though, regarding methodology and the conclusions. My comments are shown below:

• Line 96: it has been reasoned that in the massive amounts of vocal explorations …

Thanks for noticing the ungrammaticality of the sentence. We found and corrected it. The sentence now reads: “It has been reasoned that the massive amount of vocal exploration in human infants forms a foundation for language.”

METHODOLOGY/RESULTS

• I have an issue with the inclusion of the category “social plauful” in Universal vocalizations. Although animal sounds or object sounds are echopoietic, they are still conventional across languages, so they are the product of learning/associations. I would have liked to know what the percentage of this category is in Universal Vocalizations across ages, and also I suspect that the fact that there is no significant difference between gestures and vocalizations in the Universal category is due to the fact that this subcategory was wrongly included in Universal instead of conventional. When this happens the difference should become evident.

The reviewer deserves a big thank you for this comment. Table 2 had an error that misled the reviewer and would surely have misled other readers. In fact the “social playful” category for vocalization was eliminated from the Universal domain prior to our final sorting of data into categories, but we forgot to eliminate the mistaken line in Table 2. There were very few instances of animal or object sounds, but in the final analysis they were all assigned to the Conventional vocal group, because we noticed, like the reviewer, that they were at least somewhat language-specific and thus should be thought of as learned. Table 2 has been revised accordingly, eliminating the offending row. Since the data were actually sorted correctly, the relation between the amount of Universal vocalization and Universal gesture was reported correctly, in accord with the reviewer’s opinion on how the sorting should have been done.

• The fact that the two coders agree only at .8 and .65 regarding social/non-social interactions creates a problem. The percentage is very low. (line 523/524). This suggests highly ambiguous data and problematic categories. Can you please explain how you handled this problem?

The reviewer may know that kappa does not actually represent a percentage, but a special measure of agreement. The standard interpretation is that values between 0.01–0.20 represent slight agreement, 0.21–0.40 are seen as fair, 0.41– 0.60 as moderate, 0.61–0.80 as substantial, and 0.81–1.00 as almost perfect agreement. Both the kappa values are thus in the “substantial” range. Percent agreement is nowadays largely disregarded as a measure of agreement because it can be misleading if expected values in cells of a comparison are widely different, as in the present case—kappa corrects for such imbalances.

In our opinion, the most important agreement issue for our paper is the similarity of the distribution of identified categories across the two coders for the three domains (Non-social, Universal, Conventional) as portrayed in Figure 1. For gesture there were 170 items categorized with only 4 that were not agreed upon. For vocalization there were many more disagreements (73/365), but the outcome still corresponds to 80% agreement (which sounds better than .65 kappa, but that’s because kappa corrects for the imbalance of expected values). A 2x3 chi-square produced a wildly significant agreement (p < .0001) between the coders on vocalization categorization into the three domains. Whenever agreement is above chance, it is justified to conduct comparative analyses across groups of data, because if there are significant differences between the groups, there is good reason to trust them, since coder disagreement constitutes noise in the data and would produce Type II, not Type I error. In essence the more noise there is in the data, the harder it is to detect real group differences.

Finally, we hope the reviewer will notice our explanation in the main text about the primary source of the differences between coders regarding categorization of vocalizations into the domains. There is inherent difficulty in getting high agreement between coders on whether an infant vocalization is directed to an eliciting adult (whether it is non-social or social-interactive-universal) in a circumstance where the infant attention is varying moment to moment. This is not a fault of our method, but an inherent characteristic of the complexity in the actions of the infant and the parent during interaction, which involves eye contact, timing and other contextual factors, all of which are not only influencing each other but also varying moment to moment. One cannot and should not expect very high agreement among coders on all factors of judgments about infant intentions in such a circumstance. Still our agreement level justifies our analyses. 

• I cannot see any figures in the Revision 1 attachment. Seeing them in the initial submission, please add descriptions of the axes in all figures

We do not understand why it has been so difficult for all the relevant files to be made available to the reviewers. You are not the first to mention the problem. We will do whatever we can to ensure you have all the files necessary.

DISCUSSION

• I also have an issue with the discussion about what constitutes a gesture and a general utilitarian movement. Also I would like to bring into this discussion the issue of development of fine motor skills which takes time for infants, even in the first year of life. 

In other words, gestures involve fine motor movements, which take longer to develop than gross motor movements and sounds (a bit like having protophones and language sounds). So although there are movements with the hands, they are not considered gestures, as the latter take long to be produced in a controlled manner.

This suggestion has arisen in prior comments on the previous paper, as well as in presentations on our work. We have added a paragraph in the Discussion section explaining our opinion. The new paragraph reads as follows:

“It might be suggested that the higher rate of vocalization than gesture in early life can be accounted for by the fact that fine motor control of the arms, hands and fingers does not develop until later than phonatory control. But the suggestion in fact merely restates the finding rather than explaining it. Yes, the motor control of protophones occurs before the motor control of arms, but the appropriate question is why? We contend that an evolutionary explanation is needed and that the fitness signaling theory provides the necessary reasoning. In accord with the theory, it was and is because of the altriciality of infants in the human line (their slow and long-term development of head control, arm control, locomotion, and fine control of the extremities) that there has been selection pressure on them to signal their wellness effectively enough to elicit the long-term care they require because of altriciality. Phonation, according to the theory, long ago in hominin history became the target of that selection pressure, and in that way phonatory activity in humans became a precocial feature of hominins and thus laid a foundation for later developments leading to language. At a later point in evolution, complex gesture and ultimately sign language can be seen as having become, in accord with the reasoning, important secondary developments to the primary evolution of language in the vocal domain.”

The following additional reasoning might be useful as a response to the reviewer’s suggestion.

It is still not widely realized that infants are very active independently in production of phonation. As cited in the ms, even preterm infants not yet released from intensive care, produce highly differentiated phonatory patterns (Oller et al., 2019). In a sense, our earlier publication (Burkhardt-Reed et al., 2021) and those of others showing motor control of phonation proceeds faster than finger and hand motor control (that would be manifest if gesture were more prominent in early development) are the evidence that justifies the reviewer’s observation: phonatory development comes earlier than the fine motor control of gesture. The real issue is why fine motor control of phonation should develop faster than fine motor control of the hands and limbs. Notice that control of the supraglottal articulators comes later in development than phonatory control—canonical babbling, for example, begins in the second half year (Lang et al., 2019; Lohmander et al., 2020; Stoel-Gammon, 1992)—and it is not clear how to scale the rate of development of supraglottal articulation in comparison with that of fine motor control of the hands and fingers. What is eminently clear is that phonation control in humans comes earlier than either fine supraglottal articulatory control or fine motor control of the hands.

One might say that humans (in comparison with many other mammals and especially other primates) are very altricial with regard to such matters as fine finger and hand control, locomotion, head control, and posture. But they are very precocial with regard to phonation. In fact their control of phonation is not only precocial, it is largely unique among primates at any age—other primates have been found to have little if any voluntary control of phonation (Hauser, 1996). Training of any non-human primate to produce any phonatory vocalization on command has proven extremely difficult and success at any level may require years of operant conditioning (Hopkins et al., 2011). In contrast, training of supraglottal acts such as blowing raspberries or sticking the tongue out is easy. An evolutionary interpretation of these facts might be simply that the human line has been naturally selected to have not only the ability but the inclination to produce phonation and to explore it from birth. The fitness signaling theory offers an explanation as to how the human phonatory capability could have been naturally selected (Griebel & Oller, 2024; Locke, 2006). One might say that the overall altriciality of the human infant, requiring very long term provisioning and protection,was compensated for through natural selection of a way that infants could elicit long term care by vocal fitness signaling in the form of exploratory phonation. 

• A final issue concerns a point acknowledged in the limitations section – there are no data from deaf children and this suggests that we cannot know whether the patterns observed are due to an evolutionary reason or whether a special emphasis on vocalizations is the result of communicative reinforcement from parent interactions throughout the first year. So if we are going to talk about an evolutionary perspective emerging from the current data, we need to account for the fact that parents respond to vocalizations from day 1 but not to gestures. So how do we know that the current data, and preference for vocalizations over gestures, are the result of a genetic/evolutionary basis of language and not just conditioning/reinforcement for sounds instead of gestures?

Indeed, we would love to make a report on vocal and gestural development in congenitally deaf infants. But the relevant data are not available to us. 

On the conditioning suggestion, even if conditioning played a big role in early rates of gesture and vocalization (which seems doubtful, see next paragraphs), vocalization could be reinforced vastly more often than gesture simply because vocalization occurs shortly after birth at high rates while gesture is essentially absent shortly after birth. For gesture, there is virtually nothing to reinforce, while for protophones, there tend to be 4-5 per minute every waking hour (Oller et al., 2021).

But widely available data (especially since all-day recordings have been available) contradict the widespread belief that human infants vocalize largely in response to caregiver elicitation and that they learn vocalization (perhaps even phonation) through operant conditioning/parent reinforcement (Long et al., 2020; Long et al., 2019). We now know that the vast majority (more than 90%) of infant protophones are produced in the absence of any elicitation, and that they are very rarely directed toward caregivers in the natural environment of the home. There is no solid evidence that prior to the end of the first year, infants ever produce sounds on the basis of adaptation to caregiver vocalizations that are not themselves imitations by the caregivers of sounds the caregivers realize are already in the infant repertoires. The widespread belief in vocal imitation as the source of infant protophones is speculation and has no basis in fact that we know of. Experimental studies that claim to show infant vocal imitation prior to the end of the first year have been, we think, widely overinterpreted. In general what they show is that infants at various points across the first year can be experimentally induced to produce sounds already in their repertoires given vocal stimulation by sounds similar to those in their repertoires (e.g., Goldstein & Schwade, 2008; Kessen et al., 1979; Kugiumutzakis, 1999), but we know of no evidence of young infants being stimulated to produce sounds not in their endogenously developed repertoires.

So where do the infant protophones come from? They appear to be the products of endogenous exploration, overwhelmingly. When they do occur in face-to-face interaction, they show signs of adaptation to make them especially good signals of fitness (Long et al., 2022), but they do not show signs of imitation until the end of the first year, and even then only rarely. Vocabulary learning of course depends on vocal imitation, but that is a phenomenon that is extremely limited until the second year.

So, although the evidence is largely observational rather than experimental, it supports the idea that vocal development is overwhelmingly endogenous in the first year. 

Of course there is no way to fully rule out the possibility that there is a role for differential reinforcement in the dramatic difference between rates of vocalization and gesture in early life. 

If the editor thinks there should be a statement acknowledging a role for differential reinforcement cannot be completely ruled out, we will add it. 

7. PLOS authors have the option to publish the peer review history of their article (what does this mean?). If published, this will include your full peer review and any attached files.

Do you want your identity to be public for this peer review? For information about this choice, including consent withdrawal, please see our Privacy Policy.

Reviewer #1: No

Reviewer #3: Yes: Loukia Taxitari

Burkhardt-Reed, M. M., Long, H. L., Bowman, D. D., Bene, E. R., & Oller, D. K. (2021). The origin of language and relative roles of voice and gesture in early communication development. Infant Behav Dev, 65, 101648. https://doi.org/10.1016/j.infbeh.2021.101648

Goldstein, M. H., & Schwade, J. A. (2008). Social feedback to infants’ babbling facilitates rapid phonological learning. Psychological Science, 19, 515-522. 

Griebel, U., & Oller, D. K. (2024). From emotional signals to symbols. Frontiers in Pychology. https://doi.org/doi: 10.3389/fpsyg.2024.1135288 

Hauser, M. (1996). The evolution of communication. MIT. 

Hopkins, W. D., Taglialatela, J., & Leavens, D. A. (2011). Do apes have voluntary control of their vocalizations and facial expressions? In A. Vilain, J. L. Schwartz, C. 

---

## [Decision Letter · Decision Letter 2]

4 Jul 2024

Frequencies and functions of vocalizations and gestures in the second year of life

PONE-D-23-20708R2

Dear Dr. Burkhardt-Reed

We’re pleased to inform you that your manuscript has been judged scientifically suitable for publication and will be formally accepted for publication once it meets all outstanding technical requirements.

Kind regards,

Elena Theodorou

Academic Editor

PLOS ONE

Additional Editor Comments (optional):

Reviewers' comments:

Reviewer's Responses to Questions

**Comments to the Author**

1. If the authors have adequately addressed your comments raised in a previous round of review and you feel that this manuscript is now acceptable for publication, you may indicate that here to bypass the “Comments to the Author” section, enter your conflict of interest statement in the “Confidential to Editor” section, and submit your "Accept" recommendation.

Reviewer #1: All comments have been addressed

Reviewer #3: All comments have been addressed

2. Is the manuscript technically sound, and do the data support the conclusions?

Reviewer #1: Yes

Reviewer #3: Yes

3. Has the statistical analysis been performed appropriately and rigorously? 

Reviewer #1: Yes

Reviewer #3: Yes

4. Have the authors made all data underlying the findings in their manuscript fully available?

Reviewer #1: Yes

Reviewer #3: Yes

5. Is the manuscript presented in an intelligible fashion and written in standard English?

Reviewer #1: Yes

Reviewer #3: Yes

6. Review Comments to the Author

Reviewer #1: Thanks again for your positive response to my (and the others reviewers') comments. I am fully satisfied that my only outstanding comment about the figures from the previous review has been addressed.

Reviewer #3: Thank you for addressing all my comments. I really like your paper! I would still like the conditioning discussion to be part of the text, as I'm sure other people will have the same question. It's a pity to leave it out. Thank you for a quick and fruitful correspondence.

7. PLOS authors have the option to publish the peer review history of their article (what does this mean?). If published, this will include your full peer review and any attached files.

Reviewer #1: No

Reviewer #3: **Yes: **Loukia Taxitari

---

## [Editor Report · Acceptance letter]

1 Dec 2024

PONE-D-23-20708R2 

PLOS ONE

Dear Dr. Burkhardt-Reed, 

I'm pleased to inform you that your manuscript has been deemed suitable for publication in PLOS ONE. Congratulations! Your manuscript is now being handed over to our production team.

Kind regards, 

on behalf of

Dr. Elena Theodorou 

Academic Editor

PLOS ONE